# Online Learning of Neural Networks

**Amit Daniely**

The Hebre University

amit.daniely@mail.huji.ac.il

**Idan Mehalel**

The Hebrew University

idanmehalel@gmail.com

**Elchanan Mossel**

MIT

elmos@mit.edu

## Abstract

We study online learning of feedforward neural networks with the sign activation function that implement functions from the unit ball in $\mathbb{R}^d$ to a finite label set $\mathcal{Y} = \{1, \ldots, Y\}$. First, we characterize a margin condition that is sufficient and in some cases necessary for online learnability of a neural network: Every neuron in the first hidden layer classifies all instances with some margin $\gamma$ bounded away from zero. Quantitatively, we prove that for any net, the optimal mistake bound is at most approximately $\texttt{TS}(d, \gamma)$, which is the $(d, \gamma)$-*totally-separable-packing* number, a more restricted variation of the standard $(d, \gamma)$-packing number. We complement this result by constructing a net on which any learner makes $\texttt{TS}(d, \gamma)$ many mistakes. We also give a quantitative lower bound of approximately $\texttt{TS}(d, \gamma) \geq \max\{1/(\gamma\sqrt{d})^d, d\}$ when $\gamma \leq 1/2$, implying that for some nets and input sequences every learner will err for $\exp(d)$ many times, and that a dimension-free mistake bound is almost always impossible. To remedy this inevitable dependence on $d$, it is natural to seek additional natural restrictions to be placed on the network, so that the dependence on $d$ is removed. We study two such restrictions. The first is the multi-index model, in which the function computed by the net depends only on $k \ll d$ orthonormal directions. We prove a mistake bound of approximately $(1.5/\gamma)^{k+2}$ in this model. The second is the *extended margin assumption*. In this setting, we assume that *all* neurons (in all layers) in the network classify every ingoing input from previous layer with margin $\gamma$ bounded away from zero. In this model, we prove a mistake bound of approximately $(\log Y)/\gamma^{O(L)}$, where L is the depth of the network.

## 1 Introduction

We study online learning of feedforward neural networks with sign activation functions that implement functions from the unit ball in $\mathbb{R}^d$, denoted by $B(\mathbb{R}^d)$, to the label set $\mathcal{Y} = \{1, \ldots, Y\}$ where $Y \geq 2$.

In more detail, we consider the following setting. An *adversary* Adv and a *learner* Lrn are rivals in a repeated game played for some unbounded number of rounds $T$. In each round $t$, Adv sends an instance $x_t \in B(\mathbb{R}^d)$ to Lrn, and Lrn sends back a prediction[1] $\hat{y}_t \in \mathcal{Y}$. Lrn then receives the true label $y_t \in \mathcal{Y}$ from Adv, and suffers the loss $1[\hat{y}_t \neq y_t]$. The goals of Lrn and Adv are opposite: Lrn's goal is to minimize the *mistake bound* $\sum_{t \in [T]} 1[\hat{y}_t \neq y_t]$, and Adv's goal is to maximize it.

Our results and analysis focus on the *realizable setting*, where there exists an unknown *target function* $\Phi^\star \colon B(\mathbb{R}^d) \to \mathcal{Y}$ implementable by some neural network, such that $y_t = \Phi^\star(x_t)$ in every round $t$. In this setting, we denote the mistake bound of Lrn on a sequence of instances $S = x_1, x_2, \ldots, x_T$ by $\texttt{M}(\texttt{Lrn}, S) = \sum_{t \in [T]} 1[\hat{y}_t \neq \Phi^\star(x_t)]$. As we explain in the sequel, the agnostic case in which the adversary is allowed to respond with $y_t \neq \Phi^\star(x_t)$ is handled by the agnostic-to-realizable reduction

---

[1]We focus on deterministic learners.

39th Conference on Neural Information Processing Systems (NeurIPS 2025).

of [Hanneke et al., 2023a] to obtain a *regret* bound of $\tilde{O}\left(\sqrt{MT}\right)$, where $M$ is the mistake bound guaranteed when realizability is enforced, and $T$ is the number of rounds.

To the best of our knowledge, the task of online learning neural networks was not extensively explored in the literature[2]. However, we do believe that this is an important task for a variety of reasons:

1. Many learning tasks are not well captured by an i.i.d. assumption and fit well as an online learning model. This includes weather prediction, financial prediction, ad click prediction etc. Given the prominent role of neural networks in various learning and prediction tasks, it is natural to study online learning algorithms for neural networks.

2. The online learning setting is adversarial, difficult, and general. Therefore, online learning mistake bounds often have implications in other settings of learning. A non-exhaustive list of examples include PAC-learning (by using online-to-batch conversions) [Littlestone, 1989], private learning [Alon et al., 2022a], and transductive learning [Hanneke et al., 2023b, 2024].

3. This is a natural task, and as our work shows, standard and natural learning techniques reveal connections between online learning of neural networks to a known geometric quantity known as the *totally-separable-packing* number (see Section 2.1), and to the widely practically applied paradigm of *pruning* neural networks [Blalock et al., 2020, Cheng et al., 2024] (see Section 3.3).

4. Some learning tasks related to transformers use an *autoregressive* learning model, which is closely linked with online learning: The recent work [Joshi et al., 2025] shows that learning with reasonable sample complexity in an autoregressive model is impossible for some PAC-learnable function classes. However, all online learnable classes are also learnable in the autoregressive model. Our work suggests that under certain assumptions on the target network and the input, autoregressive learning of neural networks could be possible with reasonable sample complexity.

As mentioned, we focus our study on neural networks with sign activation functions. While this activation function is not used in practical applications of neural networks, there are a number of good reasons for studying online learning with this activation:

**It is simple and classic.** The sign function is simple and classic, and many classic theoretical analyzes use sign as the activation function. For example, such expressivity results can be found in [Shalev-Shwartz and Ben-David, 2014]), and sample complexity bounds may be found in [Anthony and Bartlett, 2009]. Furthermore, online algorithms for neural networks with sign activation function can be seen as generalizations of the classic perceptron algorithm [Rosenblatt, 1958] for the multi-layer case.

**It is used in binarized neural networks.** The sign activation function is widely used in *binarized neural networks* (BNNs) [Hubara et al., 2016], as such networks are restricted to having binary activations and weights. The study of BNNs is motivated by the need to deploy deep learning paradigms on low-power devices, such as mobile phones, industrial sensors, and medical equipment, where computational and energy resources are limited. Such devices often lack the infrastructure required to train standard neural networks, which typically rely on powerful hardware. Moreover, the learning tasks these devices perform are often online in nature: they make real-time predictions based on a continuous stream of incoming data.

**It implies results for other activation functions.** Beyond the motivations discussed above, we argue that our results extend, to some extent, to more common activation functions. Consider the following very simple classification network: a single input neuron and a single output neuron. In this setting, the target function is $\Phi \colon [-1, 1] \to \{\pm 1\}$. The output neuron computes a real value $r \in [-1, 1]$ based on the input, and the final prediction is given by $\mathsf{sign}(r)$. Even in this minimal setting, a learner can be forced to make an unbounded number of mistakes unless additional assumptions are made. A standard assumption is that the prediction margin is bounded away from zero. That is, $|r| \geq \gamma$ for all values $r$ computed by the output neuron, for some $\gamma > 0$. Under

---

[2]Some related work we did find is mentioned and compared to in Section 3.1.

this margin assumption, the sign function can be replaced by any activation function $\sigma$ that agrees with sign whenever $|r| \geq \gamma$, including smooth activation functions, that are more commonly used in practice. The assumptions we make throughout the paper are of a similar nature to the margin assumption described above, and analogous arguments can be used to extend our results to other activation functions. For example, see Section 3.1 where we compare some of our results to results of [Rakhlin et al., 2015], who considered more general activation functions.

## 2 Main results

The following sections assume some familiarity with standard definitions from online and neural network learning. For completeness of the main body of the paper, we provide the formal setting that we consider in the following paragraph. For other relevant definitions, the reader may refer to Section A, which introduces all the necessary background in a self-contained manner.

*Online learning* is a repeated game between a learner and an adversary. The learner's goal is to classify with minimal error a stream of instances $x_1, \ldots, x_T \in \mathcal{X}$. Each round $t$ of the game proceeds as follows.

(i) The adversary picks an instance $x_t \in \mathcal{X}$, and sends it to the learner.

(ii) The learner predicts a value $\hat{y}_t \in \mathcal{Y}$.

(iii) The adversary picks $y_t \in \mathcal{Y}$ and reveals it to the learner. The learner suffers the *loss* $\mathbb{1}[\hat{y}_t \neq h(x_t)]$.

We focus on the *realizable case*, where there exists an unknown *target function* $h \colon \mathcal{X} \to \mathcal{Y}$ taken from a known concept class $\mathcal{H}$, such that $y_t = h(x_t)$ for all $t \in [T]$. In this work, $\mathcal{H}$ is a class of functions implementable by neural networks of some architecture. In the following subsections, we describe the results proved in this work.

### 2.1 Characterization

We prove that the optimal mistake bound of learning a sequence of instances $S = x_1 \ldots, x_T \in \mathcal{X}$ labeled by a target network $\mathcal{N}^\star$ with input dimension $d$ is nearly characterized by the $(d, \gamma_1(\mathcal{N}^\star, S))$-*totally-separable packing* number (or *TS-packing* number, for short), denoted by $\mathtt{TS}(d, \gamma_1(\mathcal{N}^\star, S))$, where $\gamma_1 := \gamma_1(\mathcal{N}^\star, S)$ is defined as the minimal distance[3] between a neuron in the first hidden layer of $\mathcal{N}^\star$ to an instance in $S$. By "nearly characterized", we mean that there is an upper bound quantitatively controlled by $\mathtt{TS}(d, \gamma_1)$ for all $d, \mathcal{N}^\star, S$, and that a lower bound of $\mathtt{TS}(d, \gamma_1)$ is attained for some networks $\mathcal{N}^\star$ and sequences $S$.

For any $d, \epsilon$, $\mathtt{TS}(d, \epsilon)$ is the maximal size $T$ of a subset $\{x_1, \ldots, x_T\} \subset B(\mathbb{R}^d)$ such that for any two distinct points $x_i, x_j$ there exists a hyperplane $(\boldsymbol{w}, b) \in B(\mathbb{R}^d) \times [-1, 1]$ satisfying:

1. $(\boldsymbol{w}, b)$ linearly separates $x_i$ from $x_j$.

2. For *all* $k \in [T]$, the Euclidean distance between $(\boldsymbol{w}, b)$ and $x_k$ is at least $\epsilon$.

We prove the following bounds.

**Theorem 2.1.** *There exists a learner* Lrn *such that for any target network* $\mathcal{N}^\star$ *with input dimension* $d$ *and realizable input sequence* $S$:

$$\mathtt{M}(\mathsf{Lrn}, S) = \tilde{O}\left(\frac{\mathtt{TS}(d, \gamma_1)}{\gamma_1^2}\right).$$

*Furthermore, for any learner* Lrn*, and for any* $\varepsilon > 0, d \geq 1/\epsilon^2$*, there exists a network with input dimension* $d$ *and a realizable input sequence* $S$ *such that* $\gamma_1 \geq \varepsilon$ *and*

$$\mathtt{M}(\mathsf{Lrn}, S) = \Omega(\mathtt{TS}(d, \epsilon) + 1/\epsilon^2).$$

The upper bound is proved under the assumption that $\gamma_1$ is known to the learner in advance. This is also the case in the next two upper bounds, stated in the following two subsections (with $\gamma_1$ replaced

---

[3]Assuming all neurons' weight vectors in the first hidden layer of $\mathcal{N}^\star$ are normalized to have unit $\ell_2$-norm.

by the relevant margin definition appearing in the statement). In Section 2.4 we discuss how this assumption may be removed by a variation of the *doubling trick* [Cesa-Bianchi et al., 1997], in the cost of a polynomial degradation in the mistake bound.

We prove Theorem 2.1 in Section C. Note that the difference between the upper and lower bounds is roughly quadratic in the worst case. Furthermore, the bounds on $\mathtt{TS}(d, \epsilon)$ given in Theorem G.1 imply that when $\gamma_1$ is sufficiently small, $\mathtt{TS}(d, \gamma_1)$ is much larger than $1/\gamma_1^2$, giving tighter bounds in this case. When $\gamma_1$ is large, the dependence on $1/\gamma_1^2$ in the upper bound could be more significant, but this is not catastrophic, as $1/\gamma_1^2$ is relatively small in this case.

Note that the mistake bound demonstrates no dependence on the size of the label set $\mathcal{Y}$, which is common in multiclass online learning [Daniely et al., 2015, Brânzei and Peres, 2019, Hanneke et al., 2023a]. On the other hand, as stated in Theorem G.1, $\mathtt{TS}(d, \epsilon)$ is exponential in $d$ for small $\epsilon$, and even for $\epsilon = 1/2$ it is at least linear in $d$. This implies that to obtain dimension-free mistake bounds, we must further restrict the network and input sequence. We describe two such restricted settings we have been studying, as well as the derived results, in the following two sections.

## 2.2 Improved bound in the multi-index setting

In the *multi-index* model, we assume that the target function $\Phi^\star \colon B(\mathbb{R}^d) \to \mathcal{Y}$ calculated by the target network is restricted in the following way. There exist $k \ll d$ many *unknown* orthonormal signals $s^{(1)}, \ldots, s^{(k)} \in \mathbb{R}^d$ and a function $\phi^\star \colon B(\mathbb{R}^k) \to \mathcal{Y}$ such that for every $x \in B(\mathbb{R}^d)$, we have $\Phi^\star(x) = \phi^\star(\langle s^{(1)}, x \rangle, \ldots, \langle s^{(k)}, x \rangle)$. In simple words, while the domain of $\Phi^\star$ is $B(\mathbb{R}^d)$, the value of $\Phi^\star(x)$ is not arbitrary but depends only on an unknown $k$-dimensional projection of $x$.

The motivation of studying this setting lies in the following conjectured phenomenon: There are natural learning tasks with seemingly high-dimensional input, that in fact hides a low-dimensional structure explaining their behavior [Goldt et al., 2020]. This conjectured phenomenon might partly explain why deep learning mechanisms do well on some high-dimensional learning tasks, with low sample complexity that does not match the high-dimensional input. Consequently, this model has gained significant interest in the community, and is extensively studied in the past few years, especially in the context of stochastic optimization [Arous et al., 2021, Ba et al., 2022, Bietti et al., 2022, 2023, Damian et al., 2024, Dandi et al., 2024, Lee et al., 2024, Arnaboldi et al., 2024, Cornacchia et al., 2025].

We study online learning of neural networks in this so-called multi-index model. We prove that even though the signals $s^{(1)}, \ldots, s^{(k)}$ are unknown and $k \ll d$, it turns out that the upper bound of Theorem 2.1 holds[4] with $k$ replacing $d$, assuming a multi-index model.

**Theorem 2.2.** *In the multi-index model with $k$ many unknown signals, there exists a learner* Lrn *such that for any target network with input dimension $d$ and realizable input sequence $S$:*

$$\mathtt{M}(\mathtt{Lrn}, S) = \tilde{O}\left(\frac{\mathtt{TS}(k, \gamma_1)}{\gamma_1^2}\right).$$

We prove Theorem 2.2 in Section D. Theorem G.1 implies $\mathtt{TS}(k, \gamma_1) \leq (1.5/\gamma_1)^k$. Therefore, if $k$ is, say, some universal constant, then Theorem 2.2 implies a guaranteed mistake bound of only $\mathrm{poly}(1/\gamma_1)$.

The proof idea of Theorem 2.2 is that while the $(d, \epsilon)$-TS-packing number of the domain of $\Phi^\star$ does not change, the labeling of large packings made by $\Phi^\star$ cannot be too complicated, and in fact behaves as if the labeling was made by a function with the domain $B(\mathbb{R}^k)$.

## 2.3 Improved bound with a large margin everywhere

It is natural to study the mistake bound as a function of $\gamma := \gamma(\mathcal{N}^\star, S)$, which is the minimal margin over *all* neurons and all input instances. In more detail, for every neuron and any input instance $x$, the neuron classifies the input coming from the previous layer, and this classification also has the same natural definition of margin as in the first layer. The minimal margin $\gamma$ is the minimal margin observed in all neurons and input instances. When $\gamma$ is large, a significantly better mistake bound than of Theorem 2.1 can be proved.

---

[4]The lower bound trivially holds.

**Theorem 2.3.** *There exists a learner* Lrn, *such that for any* $d \in \mathbb{N}$, *for any target network with input dimension* $d$, *and for any realizable input sequence* $S$:

$$\mathtt{M}(\mathsf{Lrn}, S) = \tilde{O}\left(\frac{\log |\mathcal{Y}|}{\gamma^{4L+2}}\right),$$

*where* $L$ *is the depth of the network.*

We prove Theorem 2.3 in Section E. A lower bound of $\Omega(\min\{1/\gamma^2, d\})$ for some networks and input sequences is easily implied by the well-known lower bound for online linear classification. Therefore, when $L$ is small, the bound of Theorem 2.3 is fairly good.

To illustrate the significance of this result over the worst-case characterization (Theorem 2.1), let's consider a prototypical case of a network with a single hidden layer and one output neuron. In such a network, the difference between the set of neurons in the first layer considered in Theorem 2.1 and the entire set of neurons is only the output neuron. However, the bound of Theorem 2.3 in this case depends only polynomially on $1/\gamma$, and does not depend on the input dimension. Thus, it suffices that the margin in the output neuron is not extremely small for the bound of Theorem 2.3 to be much better than the bound of Theorem 2.1.

Determining whether the exponential dependence on $L$ is necessary remains open. The proof of Theorem 2.3 uses a method to significantly reduce (when $\gamma$ is large) the number of neurons in the target network, which is, to the best of our knowledge, novel. Our method relies on the celebrated *uniform convergence* theorem [Vapnik and Chervonenkis, 1971]. See Section 4.4 for more details.

## 2.4 Adaptive learning

Suppose that a learner has a mistake bound of $1/\gamma^b$ guaranteed by one of the theorems presented above, where $\gamma$ is the relevant definition of margin and $b$ is the relevant exponent. We note here that the algorithms used to prove the upper bounds described so far assume that $\gamma, b$ are known and given in advance. In Section F, we show how to remove this assumption, in the price of some polynomial degradation in the mistake bound. We stress that a standard doubling trick (which usually causes only a constant degradation in the mistake bound) is insufficient here, since there are two unknown parameters.

As a by-product, not assuming knowledge of $\gamma, b$ in fact allows us to not even know which of Theorem 2.2 or Theorem 2.3 guarantees a better mistake bound, and still achieve it, up to polynomial factors.

**Theorem 2.4.** *For some target network* $\mathcal{N}^\star$ *and input sequence* $S$, *let* $M_1, M_2$ *be the mistake bounds guaranteed by the non-adaptive algorithms providing the mistake bounds of Theorem 2.2 and Theorem 2.3, respectively. Then, there exists an algorithm, that without any prior knowledge on* $\mathcal{N}^\star$ *or* $S$ *enjoys a mistake bound of*

$$\mathtt{M}(\mathsf{Lrn}, S) = O\big((\min\{M_1, M_2\})^4\big).$$

This result is obtained by simply executing a multiclass version of the *Weighted Majority* algorithm of [Littlestone and Warmuth, 1994] (described in Section A.5.1) on the adaptive version (described in Section F) of the algorithms providing the mistake bounds of Theorem 2.2 and Theorem 2.3 as experts.

## 2.5 Agnostic learning

Our results and analysis focus on the realizable case, but can be adopted to the *agnostic* setting, where the adversary is allowed to "lie" and provide responses not perfectly matching to the target network. A different, and perhaps more common point of view on agnostic learning is that the adversary never lies, but the true labels do not match any target function. Since the expressivity of neural networks is very strong, we adopt the first point of view which is somewhat more natural in our context. That is, we assume that there exists a target network producing the labels, but the adversary changes the correct label to a different label in some of the rounds. The identity and number of rounds in which Adv tampers with the data is unknown. Our goal is to minimize the learner's *regret*, defined as

$$\mathsf{Reg}(\mathsf{Lrn}, S) = \mathbb{E}\left[\mathtt{M}(\mathsf{Lrn}, S) - \sum_{t=1}^{T} \mathbb{1}[\Phi^\star(x_t) \neq y_t]\right],$$

for any (not necessarily realizable) input sequence $S$. The expectation is taken over Lrn's randomness. In contrast to the realizable case, in the agnostic case we must allow the learner to randomize its predictions in order to achieve $o(T)$ regret [Cover, 1965]. The following regret bound is obtained by applying the agnostic-to-realizable reduction of [Hanneke et al., 2023a].

**Proposition 2.5.** *There exists a learner* Lrn*, such that for any (not necessarily realizable) input sequence $S$ of length $T \geq 2M$ that has a guaranteed mistake bound $M$ by a learner* Lrn$'$ *in the realizable case (without any labels being altered by the adversary):*

$$\mathsf{Reg}(\mathsf{Lrn}, S) = \tilde{O}\left(\sqrt{MT}\right).$$

The proof of the proposition is given by closely following the proof of Theorem 4 of [Hanneke et al., 2023a], and just replacing the Littlestone dimension with $M$.

## 3 Related work

We overview and compare to previous work generally related to online learning of neural networks in Section 3.1. Our results, especially Theorem 2.1 and Theorem 2.2 are strongly related to the TS-packing number. Finding bounds on the TS-packing number is a geometric problem which is interesting on its own right. We overview some known results related to it in Section 3.2. Our bounds also rely on the possibility to identify a small set of "important" neurons in the target net, and then use only on those neurons when learning the target function. This technique reminds us of the known "pruning" methodology which is extensively studied in the literature. We overview related work on pruning in Section 3.3.

### 3.1 Previous work on online learning of neural networks

In [Sahoo et al., 2017], an online learning algorithm for neural networks is proposed, and tested experimentally. Theoretical analysis of regret bounds for *randomized neural networks* was performed by [Chen et al., 2023, Wang et al., 2024].

Most related to our work, the work of [Rakhlin et al., 2015] gave regret bounds for online learning of neural networks when the activation is Lipschitz and the loss function is convex and Lipschitz (by joining Theorem 8 and Proposition 15 in [Rakhlin et al., 2015]). Although this is still quite different from our setting, which assumes sign activation and the $0/1$ loss (which is usually used in classification problems), we may compare their regret bound to the bound obtained in this work under the extended margin assumption. Specifically, if the depth of the network is $L$, the output is binary, and a margin of $\gamma$ is assumed for all neurons, a regret bound of $\tilde{O}\left(\sqrt{T}/\gamma^{O(L)}\right)$ is obtained by joining Theorem 2.3 and Proposition 2.5. The regret bound of [Rakhlin et al., 2015] in this setting is $\tilde{O}\left(C_\ell \sqrt{T \log d}\left(\frac{B}{\gamma}\right)^{O(L)}\right)$, where $C_\ell$ is the Lipschitz constant of the loss function, $d$ is the input dimension and $B$ is an upper bound on the 1-norm of the weight vectors. To see this, note that the actual bound given by [Rakhlin et al., 2015, Theorem 8 and Proposition 15] is $\tilde{O}\left(C_\ell \sqrt{T \log d}(B \cdot C_a)^{O(L)}\right)$ where $C_a$ is the Lipschitz constant of the activation, but the margin assumption allows us to replace sign with a $C_a$-Lipschtiz function for some $C_a \geq 1/\gamma$. Although the result of [Rakhlin et al., 2015] applies to a more general setting, our bound has a few advantages:

1. It does not depend on the input dimension nor the 1-norm of the weight vectors, which could a priori depend on the network's width.

2. It is given by an explicit algorithm rather than a minimax analysis.

3. It applies to the non-convex $0/1$ loss function. An analogue result for linear classifiers was proved in [Ben-David et al., 2009, Section 5].

4. It is given by an agnostic-to-realizable reduction (Proposition 2.5). Therefore, it is finite (independent of $T$) in the realizable case (Theorem 2.3).

### 3.2 Related geometric results

The bounds in Theorem 2.1 and Theorem 2.2 are given in terms of the TS-packing number. The investigation of totally separable packing problems in geometry literature dates back to the 40's

[Goodman and Goodman, 1945, Hadwiger, 1947], and the totally-separable notion is due to Erdős, who has made some conjectures with respect to those problems, according to [Goodman and Goodman, 1945]. The works [Tóth and Tóth, 1973, Kertész, 1988] proved bounds on the density of TS-packing of circles (2-dimensional balls) and balls (3-dimensional balls), respectively. The packings considered in those works are very similar to our TS-packings, with the main difference being that we are more interested in high dimensions, as this is the typical case when dealing with neural networks. The interested reader may refer to the recent thorough survey *on separability in discrete geometry* [Bezdek and Lángi, 2024] for more information.

### 3.3 Neural networks pruning

Pruning is a popular practical paradigm used to reduce the number of computation elements in a neural network, which is useful in practice for a variety of reasons, such as reducing infrastructure costs. There is extremely vast literature on the pruning paradigm: more than three thousand papers just between 2020 and 2024, according to [Cheng et al., 2024]. We refer the interested reader to the surveys [Blalock et al., 2020, Cheng et al., 2024] for more information and references.

The pruning method in our work is a bit different from standard, practically used pruning techniques. In standard pruning, the net is pruned and then trained again as is to recover its precision (this is sometimes called "*fine-tuning*"). In this work, we identify a (desirably small) subset of the neurons that is necessary to compute the target function calculated by the network, and then learn the target function, possibly without relying on the actual architecture of the original network.

## 4 Overview of proof techniques

In this section, we informally describe the main ideas used to prove our results. We start by describing a general approach that is common to quite a few of the proofs in this paper. A similar approach was also taken in [Khalife et al., 2024]. We think of a neural network as a pipeline with two stations:

1. The first station, implemented by the first hidden layer, partitions the unit ball to $2^\ell$ many *regions* (which some of them might be empty), where $\ell$ is the number of neurons in the first hidden layer, denoted with $\mathcal{L}$. Each region is specified by a region-specifying vector $\boldsymbol{r} := \boldsymbol{r}^{(\mathcal{L})} \in \{\pm 1\}^\ell$. That is, the region of a point $x \in B(\mathbb{R}^d)$ is specified by $\boldsymbol{r}$ if for every $i \in [\ell]$ it holds that $r_i = \mathsf{sign}(\langle \boldsymbol{v_i}, x \rangle + b_i)$, where $(\boldsymbol{v_i}, b_i)$ is the $i$'th neuron of $\mathcal{L}$.

2. The second station uses all other layers to implement some function $f \colon \{\pm 1\}^\ell \to \mathcal{Y}$.

For any point $x \in B(\mathbb{R}^d)$, we denote the region-specifying vector of $x$ by $\boldsymbol{r}(x)$. The function $\Phi^\star$ calculated by the target network is thus the composition $f \circ \boldsymbol{r}$. That is, $\Phi^\star(x) = f(\boldsymbol{r}(x))$ for all $x \in B(\mathbb{R}^d)$.

The above point of view is at the heart of the high-level strategy used to prove the mistake bounds in this paper:

1. Learn the partition of $B(\mathbb{R}^d)$ to regions.
2. Learn the label of every region.

To implement this strategy, we first describe a meta-learner that uses a multiclass version of the *Weighted Majority* algorithm of [Littlestone and Warmuth, 1994], which has good guarantees if executed with an appropriate expert class. Then, we provide specific expert classes to run the meta-learner with, for the sake of obtaining the stated mistake bounds.

Naively, the number of mistakes made when learning the partition of $B(\mathbb{R}^d)$ into regions might depend on $\ell$. Since $\ell$ might be very large, this could be a significant bottleneck of the mistake bound. Therefore, a main idea in most of the bounds is to reduce the number of neurons in the net such that only neurons which are required to properly partition $B(\mathbb{R}^d)$ to regions are considered.

**Section organization.** In Section 4.1 we overview our meta-learner for learning neural networks online. This algorithm is the main framework used to prove the mistake bounds in this paper. In Section 4.2, we describe how to use the meta-learner in order to prove the upper bound of Theorem 2.1,

and we also include a brief explanation of the lower bound. In Section 4.3 we explain how to improve the upper bound of Theorem 2.1 in the multi-index model, and in Section 4.4 we explain how to improve it when an extended margin assumption holds.

## 4.1 The Meta-learner

The meta-learner executes a multiclass version the weighted majority (WM) algorithm of [Littlestone and Warmuth, 1994]. This algorithm aggregates predictions of a class $\mathcal{E}$ of experts, to a unified prediction strategy with a mistake bound that depends logarithmically on the size of the class, and linearly on the mistake bound of a best expert. In our case, each expert from $\mathcal{E}$ implements some partition of $B(\mathbb{R}^d)$ to regions and a labeling function labeling those regions. To obtain good bounds, we need to make sure that:

1. $\mathcal{E}$ is not too large.

2. At least one expert does not make too many mistakes.

In a nutshell, we are able to make sure that both items hold in the instances of the meta-learner we use, by:

1. Showing that there are not too many possible partitions of the input sequence $S \subset B(\mathbb{R}^d)$ to different regions in $B(\mathbb{R}^d)$.

2. Making sure that every possible partition of the input sequence $S \subset B(\mathbb{R}^d)$ to regions is implemented by some expert $E$, and that the labeling function of an expert with the correct partition to regions is accurate enough as well.

The first item enables $\mathcal{E}$ to be reasonably small, and the second item is necessary so that at least one expert $E^\star$ will perform well in the task of predicting the labels of $S$. We use three different instances of the meta-learner, for three different setups: general (Theorem 2.1), multi-index (Theorem 2.2), and everywhere-margin (Theorem 2.3).

## 4.2 Quantitative general characterization

### 4.2.1 Upper bound

In Theorem 2.1 we show that the optimal mistake bound for every target net $\mathcal{N}^\star$ with input dimension $d$, and for every input sequence $S$, is not much larger than $\mathrm{TS}(d, \gamma_1)$. In order to prove this bound, we use the fact that in any partition of $B(\mathbb{R}^d)$ to regions implemented by $\mathcal{N}^\star$, at most $\mathrm{TS}(d, \gamma_1)$ regions actually contain points from $S$, otherwise $S$ induces a $(d, \gamma_1)$-TS-packing which is larger than $\mathrm{TS}(d, \gamma_1)$, and this is a contradiction. Armed with this argument, we can also prove that there exists an important set of neurons $G$ of size at most $\mathrm{TS}(d, \gamma_1)$. This allows us to construct a good enough expert class $\mathcal{E}$ to execute the meta-learner with.

### 4.2.2 Lower bound

The lower bound follows from the "two stations" point of view explained in the beginning of the section. We take a $(d, \epsilon)$-TS-packing of size $\mathrm{TS}(d, \epsilon)$ as the input sequence $S$, and show that for every $\{0, 1\}$-labeling of $S$ there exists a network $\mathcal{N}^\star$ realizing it, such that $\gamma_1 \geq \epsilon$. The neurons in the first hidden layer of $\mathcal{N}^\star$ are the hyperplanes induced by the TS-packing $S$. The second hidden layer consists of neurons determining which regions induced by the first hidden layer are labeled 0, and which are labeled 1. This implies a lower bound of $\mathrm{TS}(d, \gamma_1)$.

## 4.3 The multi-index model

The proof of the mistake bound for the multi-index model (Theorem 2.2) follows the same lines of the proof of the upper bound in the general characterization result. The main difference is that since $\Phi^\star$ in fact depends only on $k \ll d$ orthonormal directions, the arguments outlined for the general case actually hold with $k$ replacing $d$. We show that if this is not the case, we can construct a $(k, \gamma_1)$-TS-packing of size larger than $\mathrm{TS}(k, \gamma_1)$, which is of course a contradiction.

## 4.4 Large margin everywhere

In this section, we explain the proof technique of Theorem 2.3. Let us focus on the case where the target net has a single hidden layer and calculates a binary function $\Phi^\star \colon B(\mathbb{R}^d) \to \{\pm 1\}$. In this case, the extended margin $\gamma$ is the minimal margin over all neurons in the hidden layer and in the single output neuron. Recall that the margin of a neuron $(\boldsymbol{v}, b)$ is $\min_{x \in S} |\langle \boldsymbol{v}, \boldsymbol{x} \rangle + b|$, where $\boldsymbol{x}$ is the input that $(\boldsymbol{v}, b)$ receives from the previous layer when the input to the network is $x$. If $\gamma$ is large, a significantly better mistake bound can be proved, compared to the case where $\gamma_1$ (the minimal margin in the hidden layer) is large but the minimal margin of the output neuron is small. Below, we briefly explain what makes such an improvemnt possible. We use known terms and results from VC-theory. The unfamiliar reader may refer to Section A for a formal background, before reading the next paragraph.

For simplicity, in the following paragraph we assume that all neurons in $\mathcal{L}$ are homogeneous (have bias $b = 0$). For every $x \in S$, define a function $h_x \colon \mathcal{L} \to \{\pm 1\}$, given by $h_x = \mathsf{sign}(\langle x, \boldsymbol{v} \rangle)$ for every $\boldsymbol{v} \in \mathcal{L}$. Note that $|\langle x, \boldsymbol{v} \rangle| \geq \gamma$ for all $x \in S, \boldsymbol{v} \in \mathcal{L}$. Using the mistake bound of the known Perceptron algorithm [Rosenblatt, 1958, Novikoff, 1962] and the online learnability characterization of [Littlestone, 1988], this implies that the VC-dimension of the class $\mathcal{H} = \{h_x : x \in S\}$ is at most $1/\gamma^2$. Therefore, we may use the celebrated uniform convergence theorem of [Vapnik and Chervonenkis, 1971] to obtain a small "representing set" of $\mathcal{L}$ with respect to any distribution $D$ of our choice. It remains to choose $D$ in a way that communicates the result of the output neuron for any $x \in S$, using only the neurons in the representing set. We show that this is possible with a representing set of size only $\tilde{O}(1/\gamma^4)$, by choosing the probabilities of $D$ to be proportional to the weights in the weight vector of the output neuron.

To extend this result to general networks, we first extend the result to a network of arbitrary depth that calcultes a single output neuron, by applying the explanation above from the output neuron backwards, up to the first hidden layer. This is where the exponential dependence on the network's depth in Theorem 2.3 comes from. To handle any label set $\mathcal{Y}$, we use the same idea on every one of the $\log |\mathcal{Y}|$ output neurons separately. This is where the logarithmic dependence on $|\mathcal{Y}|$ in Theorem 2.3 comes from.

## 5 Open questions and future work

### 5.1 Open questions

**Quantitative gaps in mistake bounds.**    There are some quantitative gaps in the mistake bounds that it will be nice to remove.

In Theorem 2.1, there is a multiplicative gap of approximately $\min\{1/\gamma_1^2, \mathtt{TS}(d, \gamma_1)\}$ between the upper and lower bound. Can we find the correct optimal mistake bound?

Theorem 2.3 exhibits an exponential dependence on the depth of the target net. Is this dependence inevitable? For example, Khalife et al. [2024] show how to reduce the number of hidden layers in a network with sign activations to only two. Perhaps this idea could be used to improve the exponential dependence on the network's depth.

**Better adaptive algorithm.**    Our adaptive algorithm Adap (Figure 4) has a polynomially worse mistake bound than the non-adaptive learner given to it as input. This is in contrast with other online learning problems, where adaptiveness costs only a constant factor, or even less than that [Cesa-Bianchi et al., 1997, Filmus et al., 2023, 2024, Chase and Mehalel, 2024]. Does a better adaptive algorithm exist for online learning neural networks? More broadly, given an online learner with mistake bound $a^b$ on the input sequence $S$, that requires knowledge of both $a$ and $b$, is there an adaptive version of Lrn with mistake bound $\tilde{O}(a^b)$ that does not require the knowledge of neither $a$ nor $b$?

### 5.2 Future work

**Better bounds on the TS-packing number.**    We prove (Theorem 2.1) that the optimal mistake bound of learning an input sequence $S$ realizable by a neural network $\mathcal{N}^\star$ with input dimension

$d$ is controlled by $\mathtt{TS}(d, \gamma_1(\mathcal{N}^\star, S))$. However, we are not aware of any bounds on $\mathtt{TS}(d, \epsilon)$ for a high dimension $d$, except for a trivial upper bound given by an upper bound on the standard packing number, and a relatively straightforward lower bound we prove in Theorem G.1. Besides having a clear application in mistake bound analysis for neural networks, finding tighter bounds on $\mathtt{TS}(d, \epsilon)$ is an interesting problem in its own right. A similar problem was pointed out also in a recent survey on the topic [Bezdek and Lángi, 2024]. In a somewhat different direction, it would also be interesting to understand if there are efficient methods to approximate $\mathtt{TS}(d, \epsilon)$ for given $d, \epsilon$.

**Computationally efficient learning.** Our meta-learner described in Section B uses a set of more than $g^g$ experts that need to be queried and updated in every round, where $g$ is the size of the set of "important" neurons in the target net. Even in the most optimistic settings of Section D and Section E, it holds that $g \geq 1/\gamma$, which means that more than $\left(\frac{1}{\gamma}\right)^{1/\gamma}$ experts are maintained by the meta-learner. If $\gamma$ is small, this is very inefficient in terms of computation. Can we achieve good mistake bounds with computationally efficient algorithms?

**Study of other activation functions.** This work studies the sign activation function, which is basic and natural. However, modern neural networks often use other activations, such as ReLU or its variations. It will be interesting to study similar questions in the presence of more popular activations such as ReLU. We do not see how to extend our analysis to the ReLU activation, which might require a fundamentally different approach from the one taken in this work.

## Acknowledgments

The research described in this paper was funded by the European Research Council (ERC) under the European Union's Horizon 2022 research and innovation program (grant agreement No. 101041711), the Israel Science Foundation (grant number 2258/19), and the Simons Foundation (as part of the Collaboration on the Mathematical and Scientific Foundations of Deep Learning).

EM acknowledges the support of the Theoretical Foundations of Deep Learning (NSF DMS-2031883), the Vannevar Bush Faculty Fellowship ONR-N00014-20-1-2826, and the Simons Investigator Award in mathematics.

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

# Appendix Table of Contents

# A  Definitions and technical background

## A.1  Standard Notation

We use $[n] = \{1, \ldots, n\}$. We define the *sign function* for all $x \in \mathbb{R}$ as $\text{sign}(x) = 1$ if $x \geq 0$ and $\text{sign}(x) = -1$ otherwise. For a natural $d$, let $B_r(\mathbb{R}^d)$ be the ball of radius $r$ in $\mathbb{R}^d$ centered at the origin. Denote $B(\mathbb{R}^d) := B_1(\mathbb{R}^d)$. The euclidean distance between $x_1, x_2 \in \mathbb{R}^d$ is $\text{dist}(x_1, x_2)$. The $\ell_2$ norm of $x_1$ is $\|x_1\|$. The unit vector in direction $i \in [d]$ is denoted by $\boldsymbol{e_i}$. Matrices are denoted by bold capital letters like $\boldsymbol{W}$, and vectors by bold lowercase letters like $\boldsymbol{w}$. The entries are denoted by subscript indices such as $\boldsymbol{w} = w_1, \ldots, w_d$ for $\boldsymbol{w}$ of dimension $d$. $W_{i,j}$ is the value in row $i$ and column $j$ of the matrix $\boldsymbol{W}$. For two vectors $\boldsymbol{u}, \boldsymbol{v}$ of dimension $d$, their dot product is $\langle \boldsymbol{u}, \boldsymbol{v} \rangle = \sum_{i \in [d]} u_i v_i$.

## A.2  Concept classes

Let $\mathcal{X}$ be a (possibly infinite) *domain*, and $\mathcal{Y}$ be a finite label set. A pair $(x, y) \in \mathcal{X} \times \mathcal{Y}$ is called an *example*, and an element $x \in \mathcal{X}$ is called an *instance*. A function $h \colon \mathcal{X} \to \mathcal{Y}$ is called a *hypothesis* or a *concept*. A *hypothesis class*, or *concept class*, is a non-empty set $\mathcal{H} \subset \mathcal{Y}^{\mathcal{X}}$. A *labeled input sequence* of examples $\{(x_i, y_i)\}_{t=1}^T$ is said to be *realizable* by $\mathcal{H}$ if there exists $h \in \mathcal{H}$ such that $h(x_t) = y_t$ for all $1 \leq i \leq T$. We say that such $h$ is *consistent* with the labeled input sequence, or *realizes* it. An unlabeled sequence of instances $S = x_1, \ldots, x_T$ is called an *input sequence*. An input sequence $S = x_1, \ldots, x_T$ and a function $h \in \mathcal{H}$ naturally defines the realizable labeled input sequence $(x_1, h(x_1)), \ldots, (x_T, h(x_T))$.

The concept classes we will consider in this paper will usually (but not always) be all functions computable by some *neural network*, as formally defined in Section A.4.

## A.3  VC-theory and Uniform Convergence

In the proof of Theorem 2.3, we use VC-theory, and specifically the fact that VC-classes enjoy the *uniform convergence* property.

Let $\mathcal{Y} = \{\pm 1\}$, and let $\mathcal{H} \subset \mathcal{Y}^{\mathcal{X}}$ be a concept class. A set $x_1, \ldots, x_d \in \mathcal{X}$ is *shattered* by $\mathcal{H}$ if for all $y_1, \ldots, y_d \in \mathcal{Y}$, there exists $h \in \mathcal{H}$ such that $f(x_i) = y_i$ for all $i \in [d]$. The *VC-dimension* of $\mathcal{H}$, denoted by $\text{VC}(\mathcal{H})$, is defined as the maximal size of a shattered set. If there is no such maximal size, then $\text{VC}(\mathcal{H}) = \infty$. If $\text{VC}(\mathcal{H}) < \infty$, we say that $\mathcal{H}$ is a *VC-class*.

Let $D$ be a probability distribution over $\mathcal{X}$. For any $h \in \mathcal{H}$, and for any sample $S = x_1, \ldots, x_m$ of instances from $\mathcal{X}$, define

$$Q_D(h) = \mathbb{E}_{x \sim D}[h(x)], \quad \hat{Q}_S(h) = \frac{|\{x \in S : h(x) = +1\}| - |\{x \in S : h(x) = -1\}|}{|S|}.$$

We will use the uniform convergence theorem of [Vapnik and Chervonenkis, 1971].

**Theorem A.1** (Uniform convergence [Vapnik and Chervonenkis, 1971]). *We have*

$$\Pr_{S = x_1, \ldots x_m \sim D}\left[\sup_{h \in \mathcal{H}} |Q_D(h) - \hat{Q}_S(h)| > \epsilon\right] \leq 8(em/\text{VC}(\mathcal{H}))^{\text{VC}(\mathcal{H})} e^{-m\epsilon^2/32},$$

*where the notation $S = x_1, \ldots x_m \sim D$ indicates that the sample $S = x_1, \ldots x_m$ is drawn i.i.d from $D$.*

## A.4  Neural networks

Let us formally define a neural network. We mostly follow the notation of [Petersen and Zech, 2024], as described below. Let $L \in \mathbb{N}, d_0, \ldots, d_{L+1} \in \mathbb{N}$. We denote also $d = d_0$ and $d_{out} = d_{L+1}$. In this paper, a *neural network* with the *architecture* $d_0, \ldots, d_{L+1}$ is a function $\Phi : B(\mathbb{R}^{d_0}) \to \{\pm 1\}^{d_{L+1}}$ such that there exist *weight matrices* $\boldsymbol{W^{(\ell)}} \in \mathbb{R}^{d_{\ell+1} \times d_\ell}$ and *bias vectors* $\boldsymbol{b^{(\ell)}} \in \mathbb{R}^{d_{\ell+1}}$ for all $\ell \in \{0, \ldots, L\}$, for which the following holds. Let $\boldsymbol{x^0} = x$, and $\boldsymbol{x^{(\ell)}} = \text{sign}(\boldsymbol{W^{(\ell-1)}}\boldsymbol{x^{(\ell-1)}} + \boldsymbol{b^{(\ell-1)}})$ for $\ell \in [L+1]$, where the sign function is applied separately for each row. That is, $x_i^{(\ell)} = \text{sign}(\langle \boldsymbol{W^{(\ell-1,i)}}, \boldsymbol{x^{(\ell-1)}} \rangle + b_i)$ where $\boldsymbol{W^{(\ell-1,i)}}$ denotes the $i$'th row of $\boldsymbol{W^{(\ell-1)}}$. Then,

$\Phi(x) = \boldsymbol{x}^{(\boldsymbol{L+1})}$ for all $x \in \mathbb{R}^d$. For technical reasons, it will be convenient to assume that the sign function generating $\boldsymbol{x}^{(\boldsymbol{\ell})}$ is normalized (multiplicatively) by $1/\sqrt{d_\ell}$. For simplicity of notation, we will also usually assume (unless stated otherwise) that all bias vectors are the all-0 vector. Whenever this is assumed, it is clear that the arguments apply also when this is not the case.

Each row $\boldsymbol{W}^{(\boldsymbol{\ell},\boldsymbol{i})}$ in $\boldsymbol{W}^{(\boldsymbol{\ell})}$ is called a *neuron*. In many cases, we will fix $\ell$ and consider the rows of $\boldsymbol{W}^{(\boldsymbol{\ell},\boldsymbol{i})}$ as a sequence of separate neurons denoted by $\boldsymbol{w}_1, \ldots, \boldsymbol{w}_{\boldsymbol{d}_{\boldsymbol{\ell+1}}}$. We relate to those as the neurons in the $(\ell + 1)$'th hidden layer. We further assume that for every neuron $\boldsymbol{w}$: $\|\boldsymbol{w}\| = 1$. We use the notation $\mathcal{N} := \mathcal{N}(\boldsymbol{W}^{(\boldsymbol{0})}, \ldots, \boldsymbol{W}^{(\boldsymbol{L})})$ to refer to the neural network itself, as an ordered collection of weight matrices used to calculate the appropriate function $\Phi$ as described above.

## A.5 Online learning

Online learning is a repeated game between a learner and an adversary. The learner's goal is to classify with minimal error a stream of instances $x_1, \ldots, x_T \in \mathcal{X}$. Each round $t$ of the game proceeds as follows.

(i) The adversary picks an instance $x_t \in \mathcal{X}$, and sends it to the learner.

(ii) The learner predicts a value $\hat{y}_t \in \mathcal{Y}$.

(iii) The adversary picks $y_t \in \mathcal{Y}$ and reveals it to the learner. The learner suffers the *loss* $\mathbb{1}[\hat{y}_t \neq h(x_t)]$.

We focus on the *realizable case*, where there exists an unknown *target function* $h \colon \mathcal{X} \to \mathcal{Y}$ taken from a known concept class $\mathcal{H}$, such that $y_t = h(x_t)$ for all $t \in [T]$. In this work, $\mathcal{Y} = \{\pm 1\}^{d_{out}}$ and $\mathcal{H} := \mathcal{H}(d, d_{out})$ is the class of all functions $\Phi : B(\mathbb{R}^d) \to \mathcal{Y}$ implementable by a neural network $\mathcal{N}^\star$ with architecture satisfying $d_{L+1} = d_{out}$ and $d_0 = d$. We may also relate to $\mathcal{Y}$ as the set $[Y] = [2^{d_{out}}]$, where each $y \in \mathcal{Y}$ has a binary representation in $\{\pm 1\}^{d_{out}}$.

We model learners as functions $\mathsf{Lrn} \colon (\mathcal{X} \times \mathcal{Y})^* \times \mathcal{X} \to \mathcal{Y}$. The input of the learner has two parts: a *feedback sequence* $F \in (\mathcal{X} \times \mathcal{Y})^*$, and the current instance $x \in \mathcal{X}$. The feedback sequence is naturally constructed throughout the game: in the end of every round $t$, the learner appends $(x_t, y_t)$ to the feedback sequence. The prediction of $\mathsf{Lrn}$ in round $t + 1$ is then given by $\mathsf{Lrn}(F, x_{t+1})$, where $F$ is the feedback sequence gathered by the learner in rounds $1, \ldots, t$.

Given a learning rule $\mathsf{Lrn}$ and a labeled input sequence of examples $S = (x_1, y_1), \ldots, (x_T, y_T)$, we denote the number of mistakes that $\mathsf{Lrn}$ makes on $S$ by

$$\mathsf{M}(\mathsf{Lrn}; S) = \sum_{i=1}^{T} \mathbb{1}[\hat{y}_t \neq y_t].$$

This quantity is called the *mistake bound* of $\mathsf{Lrn}$ on $S$. Our goal is to design learners who minimize $\mathsf{M}(\mathsf{Lrn}; S)$ for every[5] input sequence $S$.

It is worth noting that fixing $S$ beforehand is usually linked with an *oblivious* adversary setting, in which the adversary cannot pick the examples on the fly. However, when the learner is deterministic, the adversary can simulate the entire game on its own, since we assume that the learning algorithm is known to all. Thus, oblivious and adaptive adversaries are in fact equivalent, and we will refer to the adversary as being either adaptive or oblivious, depending on whichever is more convenient in the given context.

All of our algoritms are *conservative*. Those are algorithms that change their working hypothesis only when making a mistake. Therefore, rounds where the algorithm makes a correct prediction may be ignored, and we assume that the number of rounds $T$ is exactly the number of mistakes. However, it is understood that the number of rounds may in fact be unbounded.

---

[5] In traditional online learning, one often requires a uniform bound $M$ on $\mathsf{M}(\mathsf{Lrn}; S)$ that applies to all realizable sequences $S$. This is not possible when learning neural networks, even for the simplest single-layer perceptron with input dimension 1. The interested reader may refer to [Alon et al., 2022b] for a unified theory handling with such cases.



WM($\mathcal{E}$)

**Input:** A set of experts $\mathcal{E} = \{E_1, \ldots, E_n\}$.
**Initialize:** Set $w^{(1)}(E) = 1$ for all $E \in \mathcal{E}$.
**for** $t = 1, \ldots, T$:

    1. Receive expert predictions $E_i(t) \in \mathcal{Y}$ for all $i \in [n]$.

    2. Predict
$$\hat{y}_t = \max_{y \in \mathcal{Y}} \sum_{i \in [n]:E_i(t)=y} w^{(t)}(E_i).$$

    3. Receive $y_t$.

    4. If $y_t = \hat{y}_t$: set $w^{(t+1)}(E_i) = w^{(t)}(E_i)$ for all $i \in [n]$.

    5. If $y_t \neq \hat{y}_t$: set $w^{(t+1)}(E_i) = w^{(t)}(E_i)/2$ for all $i$ so that $E_i(t) \neq y_t$, and $w^{(t+1)}(E_i) = w^{(t)}(E_i)$ for all other $i \in [n]$.



Figure 1: The multiclass weighted majority algorithm.

### A.5.1 Multiclass weighted majority

The algorithms we present use a conservative, straightforward multiclass extension of the well-known binary weighted majority (WM) algorithm of [Littlestone and Warmuth, 1994]. To the best of our knowledge, this simple extension does not appear in the literature, so we include it here for completeness, and it is described in Figure 1. The WM algorithm is executed with a family $\mathcal{E} = \{E_1, \ldots, E_n\}$ of $n$ many experts. Similarly to the standard online learning setting presented in Section A.5, the setting in which WM is executed is a repeated game between a learner and an adversary, where in the beginning of each round $t$ every expert $E_i$ gives its prediction $E_i(t) \in \mathcal{Y}$. The learner's goal is to make as few as possible prediction mistakes compared to $L$, the minimal number of mistakes made by a single expert.

The multiclass extension of the weighted majority algorithm has the same mistake bound as the standard binary version of the algorithm.

**Proposition A.2.** WM($\mathcal{E}$) *makes at most* $3(L + \log n)$ *many mistakes where $L$ is the number of mistakes made by an expert with a minimal number of mistakes, and* $n = |\mathcal{E}|$.

*Proof.* For $y \in \mathcal{Y}$, let
$$W_y^{(t)} = \sum_{i \in [n]:E_i(t)=y} w^{(t)}(E_i), \quad \text{and} \quad W^{(t)} = \sum_{y \in \mathcal{Y}} W_y^{(t)}.$$

In simple words, $W_y^{(t)}$ is the sum of weights of all experts predicting $y$ in round $t$, and $W^{(t)}$ is the sum of weights of all experts in round $t$. By the prediction rule, in any round $t$ we have $W_{y_t}^{(t)} \leq W_t/2$. Indeed, by definition of $\hat{y}_t$ we have $W_{y_t}^{(t)} \leq W_{\hat{y}_t}^{(t)}$. Therefore if $W_{y_t}^{(t)} > W_t/2$ then we have $W_{y_t}^{(t)} + W_{\hat{y}_t}^{(t)} > W_t$, which is a contradiction. Therefore, it holds that $W^{(t)} - W_{y_t}^{(t)} \geq W^{(t)}/2$. So by the update rule, we have $W^{(t+1)} \leq W_t - \frac{1}{2} \cdot W^{(t)}/2 \leq \frac{3}{4} W^{(t)}$. On the other hand, in every round $t$ we have $W^{(t)} \geq 1/2^L$. Therefore, since $W^{(1)} = n$, for any number of mistakes $T$ we have: $1/2^L \leq W^{(T)} \leq n \cdot (3/4)^T$. Solving this inequality for $T$ gives the stated upper bound. $\square$

## B  A meta online learner for neural networks

In this section, we describe a meta-learner for online learning neural networks, used to prove our upper bounds. Generally speaking, the meta-learner's main theme is to execute WM($\mathcal{E}$) on an appropriate class of experts $\mathcal{E}$.

---

**Neuron$_p$**

**Input:** $p \in \{0, 1\}^T$ .
**Initialize:** A $d$-ary all-0 vector $w$.
**for** $t = 1, \ldots, T$**:**

      1. Compute $\hat{y}_t = \text{sign}(\langle w, x_t \rangle)$.

      2. If $p_t = 1$:

          (a) If $\hat{y}_t < 0$, update $w := w + x_t$.

          (b) If $\hat{y}_t \geq 0$, update $w := w - x_t$.

      3. Return $\hat{y}_t$.

---

Figure 2: A perceptron with updates given by the "manual" vector $p$.

Let $\Phi^\star \colon B(\mathbb{R}^d) \to \mathcal{Y}$ be the target function, calculated by some neural network $\mathcal{N}^\star$ called the *target net*. Fix the input sequence $S = x_1, \ldots, x_T \in B(\mathbb{R}^d)$. For any neuron $W^{(\ell,i)}$ in $\mathcal{N}^\star$, let

$$\gamma_{W^{(\ell,i)}}(S) = \min_{x^{(\ell)} : x \in S} |\langle W^{(\ell,i)}, x^{(\ell)} \rangle|.$$

This quantity is called the *margin* of $W^{(\ell,i)}$ with respect to $S$. We assume w.l.o.g that for all $\ell, i$, $W^{(\ell,i)}$ is a *maximum margin classifier* for $S$. That is, there is no other hyperplane $W$ such that $\gamma_W(S) > \gamma_{W^{(\ell,i)}}(S)$, and both have the same sign on every input from previous layer, for all $x \in S$. In this section, we will only care about the margin of the neurons in the first hidden layer. Namely, denote

$$\gamma_1(\mathcal{N}^\star, S) = \min_{i \in [d_1]} \gamma_{W^{(0,i)}}(S).$$

When the identity of $\mathcal{N}^\star$ or $S$ (or both) is clear, we may omit them from the notation.

We may now describe the meta-learner in more detail. As mentioned, the main idea of the learner is to execute $\text{WM}(\mathcal{E})$, where $\mathcal{E}$ is chosen to be sufficiently "good". What makes a class of experts $\mathcal{E}$ "good"? In a nutshell:

1. The set $\mathcal{E}$ should not be too large.

2. At least one expert in $\mathcal{E}$ will not make too many mistakes.

If we have both guarantees with good enough numeric values, Proposition A.2 implies a good mistake bound.

How can we satisfy both guarantees? Each expert in the class $\mathcal{E}$ is an algorithm of type $\text{Expert}_{G,L_G}$ described in Figure 3, which, as suggested by its notation, is parametrized by a sequence of algorithms $G$, and a function $L_G$. The algorithms in $G$ are instances of the algorithm $\text{Neuron}_p$ described in Figure 2, who simulate a neuron, where each instance of it is parametrized by a different vector $p \in \{\pm 1\}^T$, where $T$ is the mistake bound of the meta-learner when executed with $\mathcal{E}$ (or, simply the number of rounds in the game, as WM is conservative). The binary vector $p$ functions as a "manual" for $\text{Neuron}_p$, telling which hyperplane it should converge to. This idea is inspired by the technique of [Ben-David et al., 2009].

The function $L_G$ is a labeling function from the set of regions in $B(\mathbb{R}^d)$ induced by intersections of halfspaces defined by the neurons in $G$, to $\mathcal{Y}$. The idea is therefore to use $G$ to partition $B(\mathbb{R}^d)$ to different regions, and then identify the correct label for every region. In order to satisfy the first guarantee above, we will have to show that the "correct" labeling function $L_G$ can be defined for a relatively small $G$.

### B.1 Meta mistake bound

We would now analyze a meta mistake bound to be used in our instances of the meta algorithm. We first define some additional notation, formalizing the idea of partitioning $B(\mathbb{R}^d)$ to regions based

$$\boxed{\begin{array}{l}
\qquad\qquad\qquad\qquad\qquad \mathsf{Expert}_{G,L_G}\\[4pt]
\textbf{Input: } \text{A sequence } G = (\boldsymbol{p_1}, \ldots, \boldsymbol{p_g}) \text{ of } g \text{ many vectors in } \{0,1\}^T; \text{ A labeling function}\\
L_G \colon \{\pm 1\}^g \to \mathcal{Y}.\\
\textbf{for } t = 1, \ldots, T\textbf{:}\\
\qquad 1. \text{ Construct } \boldsymbol{r} \in \{\pm 1\}^g \text{ such that for all } i:\\[4pt]
\qquad\qquad\qquad\qquad\qquad r_i = \mathsf{sign}(\langle \boldsymbol{q_i}, x_t \rangle),\\[4pt]
\qquad\quad \text{where } \boldsymbol{q_i} \text{ is the hyperplane maintained by } \mathsf{Neuron}_{\boldsymbol{p_i}}.\\
\qquad 2. \text{ Send } L_G(\boldsymbol{r}) \text{ to the meta-algorithm as the expert's prediction.}\\
\qquad 3. \text{ Retrieve } y_t \text{ from the meta-algorithm.}\\
\qquad 4. \text{ If } p_t = 0 \text{ for every neuron } \boldsymbol{p} \in G, \text{ update } L_G(\boldsymbol{r}) = y_t.
\end{array}}$$

Figure 3: An expert parametrized by a sequence of neurons.

on $G$. For a sequence $G = (\boldsymbol{w_1}, \ldots, \boldsymbol{w_g})$ of hyperplanes, we can partition the unit ball to a set of distinct *regions* (where some of them may be empty) by region-specifying vectors of the form $\{\pm 1\}^g$, where the $i$'th bit is the sign of the $i$'th hyperplane. For a region-specifying vector $\boldsymbol{r^{(G)}}$ for $G$, we denote the *region* of $\boldsymbol{r^{(G)}}$ by $R(\boldsymbol{r^{(G)}})$, which is the set of all $x \in B(\mathbb{R}^d)$ agreeing with the signs defined by $\boldsymbol{r^{(G)}}$. If the identity of the sequence of hyperplanes $G$ is clear, we omit the $(G)$ superscript. Note that for any two distinct region-specifying vectors, their appropriate regions are disjoint as they lie in different sides of at least one hyperplane. We use the notation $\boldsymbol{r}(x)$ for the region-specifying vector of $x$. That is, for all $i \in [g]$, we have $\mathsf{sign}(\langle \boldsymbol{w_i}, x \rangle) = r_i(x)$. We may also abbreviate $R(x) := R(\boldsymbol{r}(x))$. Another useful notation is $S(\boldsymbol{r}) := R(\boldsymbol{r}) \cap S$.

Fix the input sequence $S$ and let $\gamma_1 := \gamma_1(\mathcal{N}^\star, S)$. Let $G^\star = (\boldsymbol{w_1}, \ldots, \boldsymbol{w_g})$ be a sequence of neurons taken from the first hidden layer of the target net, such that there exists a function $f \colon \{\pm 1\}^g \to \mathcal{Y}$, satisfying that for every $x \in S$ it holds that $f(\boldsymbol{r}(x)) = \Phi^\star(x)$. Note that taking the entire first hidden layer must satisfy this condition, and the challenge is to find smaller sequences satisfying it. For an expert $E := \mathsf{Expert}_{G,L_G}$, denote the number of mistakes it makes on the input sequence $S$ by $M(E) = M_1(E) + M_2(E)$, where $M_1(E)$ is the number of mistakes in which the $x \in S$ misclassified by $E$ satisfies $\boldsymbol{r^{(G)}}(x) \neq \boldsymbol{r^{(G^\star)}}(x)$. That is, this is the type of mistakes that occur because the region of $x$ is not correctly identified by $E$. We call those mistakes of *the first type*. The mistakes of *the second type* counted in $M_2(E)$ are all other mistakes. Namely, for an $x$ misclassified because of a mistake of the second type, it holds that $\boldsymbol{r^{(G)}}(x) = \boldsymbol{r^{(G^\star)}}(x)$ but $L_G(x) \neq f(x)$. That is, the mistake is caused by a local incorrect choice of $L_G$. We now define the type of expert classes that we use, accompanied with two propositions showing why they are "good".

**Definition B.1.** For a number of neurons $g$ and number of rounds $T$, an expert class $\mathcal{E}$ of experts of type $\mathsf{Expert}_{G,L_G}$ is $(g, T)$-representing if for every collection $P$ of size $g$ of vectors $\boldsymbol{p} \in \{0,1\}^T$ with at most $1/\gamma_1^2$ many 1's, there exists an expert $\mathsf{Expert}_{G,L_G} \in \mathcal{E}$ such that the vectors of $G$ are precisely those of $P$.

**Proposition B.2.** *For all $g, T$ larger than a universal constant, there exists a $(g, T)$-representing class $\mathcal{E}$ such that*

$$|\mathcal{E}| \leq \left(T^{1/\gamma_1^2} + g\right)^g.$$

*Proof.* We choose $\mathcal{E}$ who satisfies the requirements of Definition B.1 of minimal size: Let $\mathcal{P} \subset \{0,1\}^T$ be the set of all $\{0,1\}$-valued vectors with at most $1/\gamma_1^2$ many 1's, and let $\mathcal{G} = \{G \subset \mathcal{P} : |G| = g\}$, where here $G \subset \mathcal{P}$ denotes a collection taken from $\mathcal{P}$ (possibly with repetitions), arbitrarily ordered as a sequence. Let $\mathcal{E} = \{\mathsf{Expert}_{G,L_G} : G \in \mathcal{G}\}$ where $L_G$ is some labeling function. Then:

$$|\mathcal{E}| \leq \left(\binom{\sum_{\leq 1/\gamma_1^2}^{T}}{g} + g\right) \leq \left(T^{1/\gamma_1^2} + g\right)^g,$$

as required. $\qquad\qquad\square$

**Proposition B.3.** *Let $g$ be the size of $G^\star$, let $T$ be the number of rounds, and let $\mathcal{E}$ be a $(g, T)$-representing class. Then there exists $\mathsf{Expert}_{G, L_G} \in \mathcal{E}$ such that:*

$$M_1(\mathsf{Expert}_{G, L_g}) \leq g/\gamma_1^2.$$

*Furthermore, in every round $t$ in which $\mathsf{Expert}_{G, L_G}$ makes a mistake, the mistake is of the first type if and only if there exists $\boldsymbol{p} \in G$ such that $p_t = 1$.*

*Proof.* Consider an execution of the known perceptron algorithm of [Rosenblatt, 1958] on the labeled input sequence $S(\boldsymbol{w_j}) = (x_1, \mathsf{sign}(\langle \boldsymbol{w_j}, x_1 \rangle)), \ldots, (x_T, \mathsf{sign}(\langle \boldsymbol{w_j}, x_T \rangle))$, for some $\boldsymbol{w_j} \in G^\star$. By the perceptron's mistake bound guarantee [Novikoff, 1962], it will make at most $1/\gamma_1^2$ many mistakes. Therefore, there exists a vector $\boldsymbol{p} \in \{0, 1\}^T$ with at most $1/\gamma_1^2$ many 1's, such that $p_t = 1$ if and only if the perceptron algorithm errs on round $t$ when executed on the input sequence $S(\boldsymbol{w_j})$. Therefore, for that $\boldsymbol{p}$, algorithm $\mathsf{Neuron}_{\boldsymbol{p}}$ classifies $S(\boldsymbol{w_j})$ correctly everywhere except for rounds $t$ with $p_t = 1$. Thus, there exists an expert $\mathsf{Expert}_{G, L_G}$ such that $G = (\boldsymbol{p_1}, \ldots, \boldsymbol{p_g})$ is a sequence of vectors in $\{0, 1\}^T$ satisfying the following: There exists a permutation $p \colon [g] \to [g]$, such that for any $j \in [g]$, $\mathsf{sign}(\langle \boldsymbol{w_{p(j)}}, x_t \rangle) = \mathsf{sign}(\langle \boldsymbol{q_j}, x_t \rangle)$ for all $t \in [T]$ except for at most $1/\gamma_1^2$, where $\boldsymbol{q_j}$ is the hyperplane maintained by $\mathsf{Neuron}_{\boldsymbol{p_j}}$. The role of the permutation $p$ is simply to match between the indices of $G^\star$ and the indices of $G$. By summing over all neurons, the total number of rounds where $\boldsymbol{r}^{(G)}(x_t) \neq \boldsymbol{r}^{(p(G^\star))}(x_t)$ is at most $g/\gamma_1^2$, where the superscript $p(G^\star)$ means that the region specifying vector's entries are according to the order induced by the permutation $p$. The same discussion implies also the "furthremore" part of the lemma. $\qquad\square$

## C   A quantitative characterization of online learning neural networks

In this section, we describe an instance of our meta-learner that has mistake bound close to optimal, when no special assumptions on the target network are assumed. Unfortunately, while this mistake bound is close to optimal, it might be very large. In the following sections, we will place further natural restrictions on the input sequence and/or the target function, and obtain better mistake bounds. We first introduce a geometric definition that will play an important role in the proved bounds.

**Definition C.1.** A $(d, \epsilon)$-*totally-separable packing*, or $(d, \epsilon)$-TS-packing, for short, is a set of distinct points $x_1, \ldots, x_T \in B(\mathbb{R}^d)$ satisfying the following. For all distinct $i, j \in [T]$ there exists a hyperplane $\boldsymbol{w} \in \mathbb{R}^d$ such that:

1. $\|\boldsymbol{w}\| = 1$.

2. $\mathsf{sign}(\langle \boldsymbol{w}, x_i \rangle) = -\mathsf{sign}(\langle \boldsymbol{w}, x_j \rangle)$.

3. $\min_{i \in [T]}\{|\langle \boldsymbol{w}, x_i \rangle|\} \geq \epsilon$.

For simplicity, we did not mention that in the formal definition, but any hyperplane is allowed to have a non-zero bias. The $(d, \varepsilon)$-*totally-separable packing number*, or $(d, \epsilon)$-TS-packing number, for short, denoted as $\mathsf{TS}(d, \epsilon)$ is the maximal number $T$ such that there exist distinct $x_1, \ldots, x_T \in B(\mathbb{R}^d)$ which form a $(d, \epsilon)$-TS-packing.

In simple words $\mathsf{TS}(d, \epsilon)$ is the maximal number of disjoint $d$-dimensional $\epsilon$-balls that can be packed in $B_{1+\epsilon}(\mathbb{R}^d)$ such that the interiors of every two balls are separated by a hyperplane that does not intersect with any of the interiors of other balls. We may now prove Theorem 2.1.

### C.1   Upper bound of Theorem 2.1

**Theorem C.2.** *There exists a learner $\mathsf{Lrn}$ such that for any target function $\Phi^\star$ computed by a target net $\mathcal{N}^\star$, and any realizable input sequence $S$:*

$$\mathtt{M}(\mathsf{Lrn}, S) = \tilde{O}\big(\mathsf{TS}(d, \gamma_1)/\gamma_1^2\big).$$

The following lemma is central in the proof of Theorem C.2.

**Lemma C.3.** *There exists a subsequence $G = (\boldsymbol{w_1}, \ldots, \boldsymbol{w_g})$ of the neurons in the first hidden layer of $\mathcal{N}^\star$, such that:*

1. $g \leq |\mathcal{Z}| \leq \mathtt{TS}(d, \gamma_1)$, where $\mathcal{Z} = \{\boldsymbol{r}^{(G)} \in \{\pm 1\}^g : S(\boldsymbol{r}) \neq \emptyset\}$.

2. There exists a function $f \colon \{\pm 1\}^g \to \mathcal{Y}$, satisfying that for every $x \in S$ it holds that $f(\boldsymbol{r}(x)) = \Phi^\star(x)$.

Before we can prove lemma C.3, we prove an auxiliary lemma. We say that $\boldsymbol{r}, \boldsymbol{r}' \in \{\pm 1\}^g$ are $j$-neighbors if $r_i = r_i' \iff i \neq j$. Let $g$ be the minimal number for which the conditions in Lemma C.3 holds. Note that $g$ is well-defined, since in the worst case it is the size of the entire collection of neurons in the first hidden layer of $\mathcal{N}^\star$.

**Lemma C.4.** *For every $j \in [g]$ there exist $j$-neighbors $\boldsymbol{r}, \boldsymbol{r}' \in \{\pm 1\}^g$ so that:*

1. *Both $S(\boldsymbol{r})$ and $S(\boldsymbol{r}')$ are non-empty.*

2. *$f(\boldsymbol{r}) \neq f(\boldsymbol{r}')$.*

*Proof.* Suppose that for some $j \in [g]$, for all $j$-neighbors $\boldsymbol{r}, \boldsymbol{r}' \in \{\pm 1\}^g$ one of the following conditions holds:

1. At least one of $S(\boldsymbol{r}), S(\boldsymbol{r}')$ is empty.

2. $f(\boldsymbol{r}) = f(\boldsymbol{r}')$.

Then, we show that we may remove $\boldsymbol{w}_j$ from $G$ and define a new function $f' \colon \{\pm 1\}^{g-1} \to \mathcal{Y}$ instead of $f$ (as defined in Lemma C.3) as follows. For $\boldsymbol{r} \in \{\pm 1\}^g$, let $\boldsymbol{r} \backslash \{j\} \in \{\pm 1\}^{g-1}$ be the vector which is identical to $\boldsymbol{r}$ without the $j$'th entry. Let $\boldsymbol{r} \in \{\pm 1\}^g$ such that $S(\boldsymbol{r}) \neq \emptyset$. Define $f'(\boldsymbol{r} \backslash \{j\}) = f(\boldsymbol{r})$. By assumption, either that $S(\boldsymbol{r}') = \emptyset$ or that $f(\boldsymbol{r}') = f(\boldsymbol{r})$, where $\boldsymbol{r}'$ is the $j$-neighbor of $\boldsymbol{r}$ and therefore using the value of $f(\boldsymbol{r})$ for the vector $\boldsymbol{r} \backslash \{j\}$ does not violate the requirements from $f$. We now only consider region-specifying vectors of length $g - 1$, and for all $x \in S$ we have $f'(\boldsymbol{r}^{G \backslash \{\boldsymbol{w}_j\}}(x)) = \Phi^\star(x)$. This contradicts the minimality of $g$. $\square$

We note that in this section we do not need the second item of Lemma C.4, but it will be useful in the following section, when proving an improved bound for the multi-index model. Let $\mathcal{Z}$ be as defined in Lemma C.4, and denote $Z = |\mathcal{Z}|$. In order to prove Lemma C.3, we will lower and upper bound $Z$, starting with the lower bound.

**Lemma C.5.** *We have $Z \geq g + 1$.*

*Proof.* By Lemma C.4, for every $j \in [g]$ there exists an unordered pair of region-specifying vectors $P_j = \{\boldsymbol{r}_{(j,+)}, \boldsymbol{r}_{(j,-)}\}$ where the $j$'th entry of $\boldsymbol{r}_{(j,+)}$ is $+1$, the $j$'th entry of $\boldsymbol{r}_{(j,-)}$ is $-1$, both agree on all other entries and both are in $\mathcal{Z}$. Therefore, it suffices to show that $|U| \geq g + 1$ where $U = \bigcup_{j \in [g]} P_j = \{\boldsymbol{r}_{(1,+)}, \boldsymbol{r}_{(1,-)}, \ldots, \boldsymbol{r}_{(g,+)}, \boldsymbol{r}_{(g,-)}\}$. Denote the distinct objects of $U$ by $\{u_1, \ldots, u_m\}$.

We define an undirected graph $Q$ with the set of vertices being $U = \{u_1, \ldots, u_m\}$ and the set of edges $P_1, \ldots, P_g$. First, note that $Q$ is a simple graph. Indeed, by definition of $Q$, for any $j$, we have $\boldsymbol{r}_{(j,+)} \neq \boldsymbol{r}_{(j,-)}$ and therefore $Q$ contains no self-loops. Furthermore, for all $i, j \in [g]$, $P_i \neq P_j$ and therefore $Q$ contains no parallel edges. We now argue that $Q$ contains no cycles and therefore is a forest. To show that fact, we argue that for any two different vertices $u, v$ such that there exists a simple path of $k$ edges connecting them, we have $\mathsf{Ham}(u, v) = k$, where $\mathsf{Ham}(u, v)$ is the hamming distance between $u, v$. Let $P_{i_1}, \ldots, P_{i_k}$ be the edges of the path. Since $Q$ is simple and the path is simple, all $i_1, \ldots, i_k$ are distinct. Therefore, precisely the bits of indices $i_1, \ldots, i_k$ are flipped between $u$ and $v$. We now use that claim to prove that $Q$ contains no (simple) cycles. Suppose that $u_1, u_2, \ldots, u_k, u_1$ is a simple cycle of length $k \geq 3$. Then, the path $u_2, \ldots, u_k, u_1$ is a simple path of length $k - 1$, and thus $\mathsf{Ham}(u_2, u_1) = k - 1$. On the other hand, that path $u_1, u_2$ is a simple path of length 1, and thus $\mathsf{Ham}(u_2, u_1) = 1$. It holds that $1 \neq k - 1$ for all $k \geq 3$, and thus we have reached a contradiction.

To conclude, $Q$ is a forest with $g$ many edges. It is well-known that the number of vertices in a forest with $g$ many edges is at least $g + 1$, proving the stated bound. As a side note, the bound holds as equality in the case where $Q$ is a tree. $\square$

**Lemma C.6.** *We have $Z \leq \mathtt{TS}(d, \gamma_1)$.*

*Proof.* It suffices to construct a $(d, \gamma_1)$-TS-packing of size $Z$. We construct such a packing as follows. Denote $\mathcal{Z} = \{r_1, \ldots r_Z\}$. We define the set $P = \{x_1, \ldots, x_Z\}$ where for all $i \in [Z]$, $x_i \in S(r_i)$ is chosen arbitrarily from the (non-empty) set $S(r_i)$. The set $P$ is well-defined by definition of $\mathcal{Z}$. Let us show that $P$ is a $(d, \gamma_1)$-TS-packing of size $Z$. It is clear that $|P| = Z$ and all points in $P$ are in $B(\mathbb{R}^d)$, since every instance is taken from a different region in $B(\mathbb{R}^d)$. To prove all other conditions, let $i, j \in [Z]$ distinct. So $x_i \in S(r_i), x_j \in S(r_j)$. Therefore, there exists an index $k$ that $r_i, r_j$ disagree on, which means that $\mathsf{sign}(\langle w_k, x_i \rangle) = -\mathsf{sign}(\langle w_k, x_j \rangle)$. In addition, $\|w_k\| = 1$ by definition of the target net, and $\min_{i \in P}\{|\langle w_k, x_i \rangle|\} \geq \gamma_1$ since $P \subset S$ and by the definition of $\gamma_1$. $\square$

*Proof of Lemma C.3.* Immediate from the jointing of Lemma C.5 and Lemma C.6. $\square$

We use a minimal size $(g, T)$-representing class $\mathcal{E}$ as defined in Definition B.1, with $g = \mathtt{TS}(d, \gamma_1)$ and $L_G \equiv 1$ for all $G$.

**Lemma C.7.** *There exists an expert $E \in \mathcal{E}$ who makes at most $2\mathtt{TS}(d, \gamma_1)/\gamma_1^2$ many mistakes.*

*Proof.* By Lemma C.3 and Proposition B.3, there exists an expert $E := \mathsf{Expert}_{G, L_G}$ such that $M_1(E) \leq \mathtt{TS}(d, \gamma_1)/\gamma_1^2$.

We now bound $M_2(E)$. Let $t$ be a round in which $E$ makes a mistake of the second type. By the "furthermore" part of Proposition B.3 we may assume that $p_t = 0$ for all $p \in G$. By definition of $\mathsf{Expert}_{G, L_G}$, in such rounds $\mathsf{Expert}_{G, L_G}$ updates $L_G(r^{(G)}) = y_t$. By definition of a mistake of the second type, we have $r^{(G)}(x_t) = r^{(p(G^\star))}(x_t)$ (where $r^{(p(G^\star))}$ is defined as in Proposition B.3), which means that $f(r^{(G)}(x_t)) = y_t$, for $f$ defined in Lemma C.3. Therefore, for any other round $t'$ such that $r^{(G)}(x_{t'}) = r^{(G)}(x_t)$ and $E$ does not make a mistake of the first type, $E$ will predict $y_t$ for $x_{t'}$, which is correct. Therefore, $E$ will make at most $Z = |\mathcal{Z}|$ mistakes of the second type, one for each region $r$ such that $S(r) \neq \emptyset$. Therefore, $M_2(E) \leq \mathtt{TS}(d, \gamma_1)$, by Lemma C.3. The total number of mistakes made by $E$ is thus at most

$$M_1(E) + M_2(E) \leq \mathtt{TS}(d, \gamma_1)/\gamma^2 + \mathtt{TS}(d, \gamma_1) \leq 2\mathtt{TS}(d, \gamma_1)/\gamma_1^2, \tag{1}$$

That completes the proof. $\square$

**Lemma C.8.** *We have*

$$M(\mathsf{WM}(\mathcal{E}), S) \leq \max\{16 \cdot \mathtt{TS}(d, \gamma_1)/\gamma_1^2 \cdot \log(\mathtt{TS}(d, \gamma_1)/\gamma_1^2), C\},$$

*where $C$ is a universal constant.*

*Proof.* First, by Proposition B.2 we have

$$|\mathcal{E}| \leq (T^{1/\gamma^2} + \mathtt{TS}(d, \gamma_1))^{\mathtt{TS}(d, \gamma_1)}.$$

By Lemma C.7, there exists an expert who makes at most $2\mathtt{TS}(d, \gamma_1)/\gamma_1^2$ many mistkaes. By the mistake bound of $\mathsf{WM}(\mathcal{E})$ in Proposition A.2, we have:

$$T \leq 3\left(\frac{2\mathtt{TS}(d, \gamma_1)}{\gamma_1^2} + \frac{\mathtt{TS}(d, \gamma_1)}{\gamma_1^2}\log T + \mathtt{TS}(d, \gamma_1)\log\mathtt{TS}(d, \gamma_1)\right).$$

For any $T > 15 \cdot \mathtt{TS}(d, \gamma_1)/\gamma_1^2 \cdot \log(\mathtt{TS}(d, \gamma_1)/\gamma_1^2)$, the above inequality is a contradiction if $\mathtt{TS}(d, \gamma_1)/\gamma_1^2$ is larger than a universal constant. Othrewise, the above inequality is a contradiction for any $T$ larger than a universal constant. This proves the stated bound. $\square$

*Proof of Theorem C.2.* Immediate from Lemma C.8. $\square$

## C.2 Lower bound of Theorem 2.1

The idea is similar to universality and expressivity results for neural networks (see, e.g., [Shalev-Shwartz and Ben-David, 2014]). Concretely, the lower bound relies on showing that every binary function on a $(d, \gamma_1)$-TS-packing can be expressed without violating the minimal $\gamma_1$ margin constraint in the first hidden layer of the target net.

**Theorem C.9.** *For any learner* Lrn, *and for any* $\varepsilon > 0, d \geq 1/\varepsilon^2$, *there exists a network with input dimension* $d$ *and a realizable input sequence* $S$ *such that* $\gamma_1 \geq \varepsilon$ *and*

$$\mathtt{M}(\mathtt{Lrn}, S) = \Omega(\mathtt{TS}(d, \epsilon) + 1/\epsilon^2).$$

*Proof.* The dependence on $1/\epsilon^2$ is implied by the known lower bound showing that the Perceptron algorithm is optimal.

Let us concentrate in the dependence on $\mathtt{TS}(d, \epsilon)$. Let $x_1, \ldots, x_T$ be a maximal $(d, \epsilon)-$TS-packing. For simplicity of notation, we prove the lower bound for the case where all induced separating hyperplanes for $x_1, \ldots, x_T$ are homogeneous, and the same argument works when this is not the case. The adversary uses $S = x_1, \ldots, x_T$ as the input sequence of instances, and forces a mistake on the learner in every round. It remains to show that for every binary labeling $y_1, \ldots, y_T \in \{\pm 1\}$ of $S$ there exists a network $\mathcal{N}^\star$ of input dimension $d$ who realizes it with $\gamma_1 \geq \epsilon$. Let $G = (\boldsymbol{w}^{(1)}, \ldots, \boldsymbol{w}^{(g)})$ be a sequence of separating hyperplanes induced by the TS-packing $S$, and let $y_1, \ldots, y_T \in \{\pm 1\}$ be a binary labeling of $S$. We construct a net $\mathcal{N}^\star$ with two hidden layers as follows. The first hidden layer consists of all hyperplanes in $G$. The second hidden layer is constructed as follows. Let $f : \{\pm\}^g \to \{\pm 1\}$ be a binary function such that for every $x_t \in S$, $f(\boldsymbol{r}^{(G)}(x)) = y_t$. The neurons in the second hidden layer are all $\boldsymbol{r} \in \{\pm 1\}^g$ such that $f(\boldsymbol{r}) = 1$, multiplicatively normalized by $1/\sqrt{g}$, and with an added bias of $\frac{1}{2g} - 1$. The output layer consists of a single output neuron with all weights being $1/\sqrt{d_2}$ (recall that $d_2$ is the width of the second hidden layer). The bias of the output neuron is $1 - 1/d_2$.

Clearly, $\gamma_1 \geq \epsilon$. It remains to show that $\Phi^\star(x_t) = y_t$ for all $x_t \in S$, where $\Phi^\star$ is the function computed by $\mathcal{N}^\star$. Let $x_t \in S$, and let us calculate $\Phi^\star(x_t)$. First note that the input for the second hidden layer is precisely $\frac{1}{\sqrt{g}}\boldsymbol{r}(x_t)$. Now, for every neuron $(\boldsymbol{q}(\boldsymbol{r}), \frac{1}{2g} - 1)$ in the second hidden layer added for some $\boldsymbol{r} \in \{\pm 1\}^g$ in the first hidden layer, we have

$$\mathsf{sign}\left(\left\langle \boldsymbol{q}(\boldsymbol{r}), \frac{1}{\sqrt{g}}\boldsymbol{r}(x_t)\right\rangle - 1 + \frac{1}{2g}\right) = \begin{cases} 1 & \boldsymbol{r} = \boldsymbol{r}(x_t), \\ -1 & \boldsymbol{r} \neq \boldsymbol{r}(x_t). \end{cases}$$

Therefore, if $y_t = 0$, then the output of every neuron in the second hidden layer is $-1/\sqrt{d_2}$ and therefore the output neuron will get the value

$$\mathsf{sign}\left(-d_2 \frac{1}{\sqrt{d_2}} \cdot \frac{1}{\sqrt{d_2}} + 1 - \frac{1}{d_2}\right) = \mathsf{sign}\left(-\frac{1}{d_2}\right) = -1.$$

Otherwise, if $y_t = 1$, then the output of every neuron in the second hidden layer is $-1/\sqrt{d_2}$, except for the neuron with weights $\boldsymbol{q}(\boldsymbol{r}(x))$ which exists by the definition of the network. Therefore the output neuron will get the value

$$\mathsf{sign}\left(-(d_2 - 1)\frac{1}{\sqrt{d_2}} \cdot \frac{1}{\sqrt{d_2}} + \frac{1}{\sqrt{d_2}} \cdot \frac{1}{\sqrt{d_2}} + 1 - \frac{1}{d_2}\right) = \mathsf{sign}\left(\frac{1}{d_2}\right) = 1,$$

completeing the proof. $\qquad\square$

Applying the lower bound on $\mathtt{TS}(d, \epsilon)$ from Theorem G.1 to the lower bound on the mistake bound proved above shows that the mistake bound can be exponential in $d$ for small $\gamma_1$, and linear in $d$ even for constant $\gamma_1$. This inevitable dependence on $d$ requires further assumptions on the target net and the input sequence in order to get dimension-free mistake bounds. We study such assumptions in the next sections.

# D   The multi-index model

In this section, we prove Theorem 2.2. For that matter, we prove variations of the lemmas proved in Section C, but with $d$ replaced by $k$. The core idea allowing this is the fact that a labeling made by $\phi^\star$ (as defined in Section 2.2) for a $(d, \epsilon)$-TS-packing induces a labeling of a $(k, \epsilon)$-TS-packing, and thus its level of complication depends on $k$ rather than on $d$.

We begin by stating an appropriate version of Lemma C.3.

**Lemma D.1.** *There exists a subsequence $G = (\boldsymbol{w_1}, \ldots, \boldsymbol{w_g})$ of the hidden neurons (hyperplanes) in the first hidden layer of $\mathcal{N}^\star$, such that:*

> 1. $g \leq |\mathcal{Z}| \leq \mathtt{TS}(k, \gamma_1)$, *where $\mathcal{Z} = \{\boldsymbol{r}^{(\boldsymbol{G})} \in \{\pm 1\}^g : S(\boldsymbol{r}) \neq \emptyset\}$.*
>
> 2. *There exists a function $f \colon \{\pm 1\}^g \to \mathcal{Y}$, satisfying that for every $x \in S$ it holds that $f(\boldsymbol{r}(x)) = \Phi^\star(x)$.*

As in Section C, we consider $g$ as the minimal size of a subsequence of the neurons in the first hidden layer for which the conditions in Lemma D.1 hold. Therefore, Lemma C.4 and Lemma C.5 can be used as is in also the multi-index setting. The proof of the improved upper bound is mainly due to the following lemma.

**Lemma D.2.** *We have $Z \leq \mathtt{TS}(k, \gamma_1)$.*

To prove this lemma, we need an additional crucial property of the neurons in $G$, proved in the lemma below. For simplicity and convenience of proof, we assume w.l.o.g until the rest of the section that $\boldsymbol{s}^{(i)} = \boldsymbol{e_i}$ for all $i \in [k]$. Indeed, if this is not the case, we can rotate the entire system to this state.

**Lemma D.3.** *For any $\boldsymbol{w} \in G$, $w_i = 0$ for all $i > k$.*

*Proof.* Suppose towards contradiction that $w_{k+1} \neq 0$ for some $\boldsymbol{w} \in G$. Suppose w.l.o.g that the index of $\boldsymbol{w}$ in $G$ is 1. By Lemma C.4, there exist $\boldsymbol{r}, \boldsymbol{r}' \in \{\pm 1\}^g$ such that $r_1 = 1, r_1' = -1$, and $r_i = r_i'$ for all $i \neq 1$, and furthermore:

> 1. $S(\boldsymbol{r}), S(\boldsymbol{r}')$ are both non-empty.
>
> 2. $f(\boldsymbol{r}) \neq f(\boldsymbol{r}')$ (for $f$ defined in Lemma D.1).

Therefore, also $R(\boldsymbol{r}), R(\boldsymbol{r}')$ are both non-empty. Therefore, there exists $\epsilon > 0$ so that there exists a ball $B$ with the following properties:

> 1. $\boldsymbol{w}$ intersects with the center of $B$.
>
> 2. $B$ has radius $\epsilon$.
>
> 3. $B \subset R(\boldsymbol{r}) \cup R(\boldsymbol{r}')$.

Let $c$ be the center of $B$. Note that $c$ has the following properties:

> 1. $\langle \boldsymbol{w}, c \rangle = 0$.
>
> 2. $\|c\| \leq 1 - \epsilon$.

Assume w.l.o.g that $w_{k+1} > 0$. Consider two points $c(\boldsymbol{r}), c(\boldsymbol{r}')$. The point $c(\boldsymbol{r})$ is identical to $c$ except that $c(\boldsymbol{r})_{k+1} = c_{k+1} + \epsilon/2$. The point $c(\boldsymbol{r}')$ is identical to $c$ except that $c(\boldsymbol{r}')_{k+1} = c_{k+1} - \epsilon/2$. Note that $c(\boldsymbol{r}), c(\boldsymbol{r}') \in B$ and furthermore that

$$\langle \boldsymbol{w}, c(\boldsymbol{r}) \rangle > 0, \quad \langle \boldsymbol{w}, c(\boldsymbol{r}') \rangle < 0,$$

since $\langle \boldsymbol{w}, c \rangle = 0$. Therefore, we have $c(\boldsymbol{r}) \in R(\boldsymbol{r})$ and $c(\boldsymbol{r}') \in R(\boldsymbol{r}')$, and since $f(\boldsymbol{r}) \neq f(\boldsymbol{r}')$ we have

$$\Phi^\star(c(\boldsymbol{r})) \neq \Phi^\star(c(\boldsymbol{r}')). \tag{2}$$

On the other hand, $c(r)_i = c(r')_i$ for all $i \leq k$. Therefore, by the multi-index assumption

$$\Phi^\star(c(\boldsymbol{r})) = \phi^\star(c(\boldsymbol{r})) = \phi^\star(c(\boldsymbol{r}')) = \Phi^\star(c(\boldsymbol{r}')),$$

which contradicts (2). $\qquad\square$

We may now prove Lemma D.2.

*Proof of Lemma D.2.* It suffices to construct a $(k, \gamma_1)$-TS-packing of size $Z$. Denote $\mathcal{Z} = \{\boldsymbol{r_1}, \dots, \boldsymbol{r_Z}\}$. Let $P' = \{x_1', \dots, x_Z'\}$, where for all $i \in [Z]$, $x_i'$ is chosen arbitrarily from the non-empty set $S(\boldsymbol{r_i})$. Let $P = \{x_1, \dots, x_Z\}$, where for every $i \in [Z]$, let $x_i$ be as $x_i'$, but only with its first $k$ indices. Let us show that $P = \{x_1, \dots, x_Z\}$ is a $(k, \gamma_1)$-TS-packing of size $Z$. First, for all $i$, $x_i' \in B(\mathbb{R}^d)$, and therefore $x_i \in B(\mathbb{R}^k)$. To show that $|P| = Z$, we need to show that $x_i \neq x_j$ for all distinct $i, j$. Indeed, since $x_i', x_j'$ are taken from distinct $S(\boldsymbol{r_i}), S(\boldsymbol{r_j})$, there exists $\boldsymbol{w} \in G$ so that $\mathsf{sign}(\langle \boldsymbol{w}, x_i' \rangle) \neq \mathsf{sign}\langle \boldsymbol{w}, x_j' \rangle$. Therefore, $x_i', x_j'$ must differ in at least one index where $\boldsymbol{w}$ is not zeroed. Recall that $w_t = 0$ for all $t > k$ by Lemma D.3, and thus they differ in some index smaller than $k + 1$, and therefore $x_i \neq x_j$. So far, we have established that $P \subset B(\mathbb{R}^k)$ and that its size is $Z$. It remains to show that total separability holds with the separation parameter $\gamma_1$. We define $G_{\leq k}$ to be the same as $G$, only that all indices larger than $k$ are removed from each vector in $G$ which is then scaled accordingly to have norm 1, making them all $k$-dimensional. For $\boldsymbol{w} \in G$, denote the appropriate trimmed vector in $G_{\leq k}$ by $\boldsymbol{w}_{\leq k}$. Let $i, j \in [Z]$ be distinct indices. From the same argument already given, there exists $\boldsymbol{w}_{\leq k} \in G_{\leq k}$ so that $\mathsf{sign}(\langle \boldsymbol{w}_{\leq k}, x_i \rangle) \neq \mathsf{sign}\langle \boldsymbol{w}_{\leq k}, x_j \rangle$. Finally, we claim that $\min_{i \in P} |\langle \boldsymbol{w}_{\leq k}, x_i \rangle| \geq \gamma_1$. Indeed, since $P' \subset S$ we have $\min_{i \in P'} |\langle \boldsymbol{w}, x_i' \rangle| \geq \gamma_1$, which proves the claim. $\square$

*Proof of Lemma D.1.* Immediate from the joint of Lemma C.5 and Lemma D.2. $\square$

**Lemma D.4.** *There exists an expert who makes at most $2\mathsf{TS}(k, \gamma_1)/\gamma_1^2$ many mistakes.*

*Proof.* We apply the exact same arguments as in the proof of Lemma C.7, with the only difference of applying Lemma D.1 instead of Lemma C.3 in (1). $\square$

**Lemma D.5.** *We have*

$$M(\mathsf{WM}(\mathcal{E}), S) \leq \max\{16 \cdot \mathsf{TS}(k, \gamma_1)/\gamma_1^2 \cdot \log(\mathsf{TS}(k, \gamma_1)/\gamma_1^2), C\},$$

*where $C$ is a universal constant.*

*Proof.* We apply the same arguments used in the proof of Lemma C.8, with only replacing $\mathsf{TS}(d, \gamma_1)$ with $\mathsf{TS}(k, \gamma_1)$, and Lemma C.7 with Lemma D.4. $\square$

*Proof of Theorem 2.2.* Immediate from Lemma D.5. $\square$

# E   Learning with large margin everywhere

Fix the input sequence $S \subset (\mathbb{R}^d \times \mathcal{Y})^*$. Recall the definition of a neuron's margin from Section C. For any neuron $\boldsymbol{W}^{(\ell,i)}$ in $\mathcal{N}^\star$, let

$$\gamma_{\boldsymbol{W}^{(\ell,i)}}(S) = \min_{\boldsymbol{x}^{(\ell)}:x \in S} |\langle \boldsymbol{W}^{(\ell,i)}, \boldsymbol{x}^{(\ell)} \rangle|.$$

We now define the minimal margin in the entire net:

$$\gamma(\mathcal{N}^\star, S) = \min_{\ell \in \{0, \dots, L\}} \min_{i \in [d_{j+1}]} \gamma_{\boldsymbol{W}^{(\ell,i)}}(S).$$

When the identity of $\mathcal{N}^\star$ or $S$ (or both) is clear, we may omit them from the notation.

We will now prove Theorem 2.3. In section E.1, we explain how to prune the network based on its minimal margin in the case of a network that implements binary classification and has a single hidden layer. Then, in Section E.2 we explain how to extend the technique to the general case. Finally, in Section E.3 we give the learning algorithm itself.

## E.1 Margin-based pruning with a single hidden layer and two labels

Let $\mathcal{N}^\star$ be the target net, and suppose that $\mathcal{N}^\star$ has a single hidden layer of width $\ell$, and that it implements a binary function $\Phi^\star \colon B(\mathbb{R}^d) \to \{\pm 1\}$ (and therefore has a single output neuron). The collection of hidden neurons is denoted by $\mathcal{L} = (v_1 \dots, v_\ell)$.

The main idea allowing the mistake bound of Theorem 2.3 is that the hidden layer of $\mathcal{N}^\star$ in fact contains only $\tilde{O}(1/\gamma^4)$ "important" neurons. As usual, by "important", we mean that for every $x \in S$, $\Phi^\star(x)$ depends only on their output. This is formalized and proved in the following lemma, via uniform convergence.

**Lemma E.1.** *There exists a sequence $G = (w_1, \dots w_g)$ taken from the $\ell$ neurons in the hidden layer of $\mathcal{N}^\star$ such that:*

1. $g = \tilde{O}(1/\gamma^4)$.

2. *For all $x \in S$:*

$$\Phi^\star(x) = \operatorname{sign}\left( \sum_{i=1}^{g} \operatorname{sign}(\langle w_i, x \rangle) \right).$$

*Proof.* We define a binary hypothesis class $\mathcal{H}$ as follows. The domain of instances $\mathcal{X}$ is all $\ell$ neurons in the hidden layer of $\mathcal{N}^\star$. The input instances $S = x_1, \dots, x_T$ define the hypothesis class $\mathcal{H} = \{h_{x_1}, \dots, h_{x_T}\}$ in the natural way: For an instance $x \in S$ and a neuron $v \in \mathcal{X}$, $h_x(v) = \operatorname{sign}(\langle x, v \rangle)$. By assumption, $|\langle x, v \rangle| \geq \gamma$ for all $v \in \mathcal{X}$, $x \in S$. Therefore, the online perceptron algorithm will make at most $1/\gamma^2$ many mistakes on any input sequence realizable by $\mathcal{H}$. It is known, for example by [Littlestone, 1988], that a uniform mistake bound in the standard online setting for all realizable sequences upper bounds the VC-dimension of the class. Therefore $\operatorname{VC}(\mathcal{H}) \leq 1/\gamma^2$.

We now define a distribution $D$ over $\mathcal{X} = \{v_1, \dots, v_\ell\}$ (the hidden neurons). Let $o$ be the output neuron of $\mathcal{N}^\star$, and suppose w.l.o.g that all entries of $o$ are non-negative. We define $D$ to be proportional to the weights of $o$. Namely, for every $i \in [\ell]$ define

$$D(v_i) = o_i/\sqrt{\ell}.$$

Let $r(x) \in \{\pm 1\}^\ell$ be the output value vector of the hidden neurons when the input instance is $x \in S$, and let $u(r)$ be the real value calculated by the output neuron when $r$ is the output of the hidden neurons. The reason we chose $D$ as the distribution above, is that $Q_D(x)$ (as defined in Section A.3) is precisely $u(r(x))$ for any $x \in S$:

$$
\begin{aligned}
u(r(x)) &= \sum_{i \in [\ell]} o_i \cdot \frac{1}{\sqrt{\ell}} r_i(x) \\
&= \sum_{i \in [\ell]} D(v_i) \cdot r_i(x) \\
&= \sum_{i \in [\ell]} D(v_i) \cdot \operatorname{sign}(\langle v_i, x \rangle) \\
&= \mathbb{E}_{v_i \sim D}[\operatorname{sign}(\langle v_i, x \rangle)] \\
&= \mathbb{E}_{v_i \sim D}[h_x(v_i)] \\
&= Q_D(x).
\end{aligned}
$$

Theorem A.1 implies that if we draw an i.i.d sequence $G$ of $g = \left\lceil 1000 \frac{\operatorname{VC}(\mathcal{H})}{\gamma^2} \log\left( \frac{\operatorname{VC}(\mathcal{H})}{\gamma^2} \right) \right\rceil$ many neurons (possibly with repetitions) from $D$, we have

$$\Pr_{G = w_1, \dots w_g \sim D}\left[ \sup_{h_x \in \mathcal{H}} |Q_D(h_x) - \hat{Q}_G(h_x)| > \gamma/2 \right] \leq 3/4.$$

Therefore, there exists a sequence $G$ of size $g$ of hidden neurons such that

$$\sup_{h_x \in \mathcal{H}} |Q_D(h_x) - \hat{Q}_G(h_x)| \leq \gamma/2. \tag{3}$$

Note also that $\Phi^\star(x) = \text{sign}(u(\boldsymbol{r}(x)))$. Therefore, for all $x \in S$ we have

$$
\begin{aligned}
\Phi^\star(x) &= \text{sign}(u(\boldsymbol{r}(x))) \\
&= \text{sign}(Q_D(h_x)) \\
&= \text{sign}(\hat{Q}_G(h_x)) \\
&= \text{sign}\left( \sum_{i=1}^{g} \text{sign}(\langle \boldsymbol{w_i}, x \rangle) \right),
\end{aligned}
$$

where the third equality is due to (3) and the margin assumption: By the margin assumption and the second equlaity, we have $|Q_D(h_x)| = |u(\boldsymbol{r}(x))| \geq \gamma$, and therefore for any $q \in [Q_D(h_x) - \gamma/2, Q_D(h_x) + \gamma/2]$, it holds that $\text{sign}(q) = \text{sign}(Q_D(h_x))$. Equation 3 implies that $\hat{Q}_G(h_x) \in [Q_D(h_x) - \gamma/2, Q_D(h_x) + \gamma/2]$. Recall that $\text{VC}(\mathcal{H}) = 1/\gamma^2$ and thus $g = \tilde{O}(1/\gamma^4)$. That completes the proof. □

### E.2 Margin-based Pruning in the general case

We now adapt the approach from previous section that handled shallow networks and binary output to the general case. The main building block we use to extend the pruning result from the previous section to general networks is the function $f_{g,L} \colon \{\pm 1\}^{g^L} \to \{\pm 1\}$, where $g, L \in \mathbb{N}$, which we define inductively as follows. For $i \in \{0, \dots, L-1\}$, let $f_{g,L}^{(i)} \colon \{\pm 1\}^{g^{L-i}} \to \{\pm 1\}^{g^{L-(i+1)}}$ be defined as follows. Let $\boldsymbol{r} \in \{\pm 1\}^{g^{L-i}}$. For any index $j$ of the output $f_{g,L}^{(i)}(\boldsymbol{r})$:

$$
f_{g,L}^{(i)}(\boldsymbol{r})_j = \text{sign}\left( \sum_{i=g\cdot(j-1)+1}^{g\cdot j} r_i \right).
$$

In simple words, $f_{g,L}^{(i)}(\boldsymbol{r})$ takes every consecutive $g$ entries in $\boldsymbol{r}$ and uses the sign of their majority as a new entry in the output vector.

We now show how to use $f_{g,L}$ to extend the result from the previous section to the case of a general network. Suppose that the target network $\mathcal{N}^\star$ has $L$ hidden layers.

**Lemma E.2.** *Fix an output neuron in $\mathcal{N}^\star$, and let $o(x) \in \{\pm 1\}$ be its value when $x \in S$ is the input. Let $g$ be as in the proof of Lemma E.1. Then there exists a sequence $G$ of $g^L$ neurons from the first hidden layer of $\mathcal{N}^\star$, such that:*

$$
f_{g,L}(\boldsymbol{r}(x)) = o(x)
$$

*for all $x \in S$, where $\boldsymbol{r}(x)$ is, as usual, the output vector of the neurons in $G$.*

*Proof.* The proof is by repeatedly applying Lemma E.1 from the output neuron backwards. Lemma E.1 immediately implies that in the $L$'th hidden layer, there exists a sequence $G_L = (\boldsymbol{w_1^{(L)}}, \dots, \boldsymbol{w_g^{(L)}})$ of $g$ many neurons so that the sign of the sum of their outputs when $x$ is the input gives $o(x)$ for any $x \in S$. Now, since $\gamma$ is the minimal margin in the entire net, we can relate to each $\boldsymbol{w_i^{(L)}}$ as an output neuron, and again by Lemma E.1, in the $(L-1)$'th hidden layer, there exists a sequence $G_{L-1} = (\boldsymbol{w_1^{(L-1)}}, \dots, \boldsymbol{w_g^{(L-1)}})$ of $g$ many neurons so that the sign of the sum of their outputs when $x$ is the input equals to the output of neuron $\boldsymbol{w_i^{(L)}}$ when $x$ is the input, for any $x \in S$. Repeating the argument all the way up to the first layer gives the stated result. □

Lemma E.2 implies that a sequence of $\tilde{O}(1/\gamma^{4L})$ neurons from the first hidden layer suffices to calculate the output of a single output neuron. Therefore, it is clear that $\tilde{O}\left( \frac{\log |\mathcal{Y}|}{\gamma^{4L}} \right)$ many neurons suffice to calculate the output of all output neurons. It remains to learn those $\tilde{O}\left( \frac{\log |\mathcal{Y}|}{\gamma^{4L}} \right)$ neurons, and then just use $f_{g,L}$ to calculate the output neurons. This is done by our meta-learner from Section B.

### E.3 Learning algorithm with margin everywhere

As mentioned earlier, since there are $\lceil \log |\mathcal{Y}| \rceil$ output neurons, Lemma E.2 implies that $\tilde{O}\left(\frac{\log |\mathcal{Y}|}{\gamma^{4L}}\right)$ many neurons are suffice to calculate the output of all output neurons by calculating $f_{g',L}(\boldsymbol{r}^{(G)}(x))$ for every sequence $G$ of size $g' = \tilde{O}(1/\gamma^{4L})$ of neurons that suffice to calculate the output of a single output neuron. So we use our meta-learner with a $(g, T)$-representing class $\mathcal{E}$ of minimal size, with $g = C \cdot \frac{\log |\mathcal{Y}|}{\gamma^{4L}} \log(1/\gamma^L)$, where $C$ is some large enough universal constant, and $L_G$ being the function that calculates every output neuron separately by the value of $f_{g',L}$ on the appropriate sequence of neurons given in Lemma E.2.

**Lemma E.3.** *We have*
$$\mathrm{M}(\mathrm{WM}(\mathcal{E}), S) = \tilde{O}\left(\frac{\log |\mathcal{Y}|}{\gamma^{4L+2}(S)}\right).$$

*Proof.* By Proposition B.2, we have
$$|\mathcal{E}| \leq (T^{1/\gamma^2} + g)^g$$

where $g = C \cdot \frac{\log |\mathcal{Y}|}{\gamma^{4L}} \log(1/\gamma^L)$. By Lemma E.2, there exists an expert $E \in \mathcal{E}$ for which $M_2(E) = 0$ and thus $M(E) \leq g/\gamma^2$. Therefore, the mistake bound guarantee of WM implies that

$$T \leq 3\left(\frac{g}{\gamma^2} + \frac{g}{\gamma^2}\log T + g\log g\right).$$

The above inequality is a contradiction for any

$$T > 9C\frac{\log |\mathcal{Y}|}{\gamma^{4L+2}}\log\left(\frac{\log |\mathcal{Y}|}{\gamma^L}\right),$$

which proves the stated bound. $\qquad\square$

*Proof of Theorem 2.3.* Immediate from Lemma E.3. $\qquad\square$

## F  Adapting to the correct parameters

Our mistake bounds are of the form $1/\gamma^b$, where $\gamma < 1$ is the relevant definition of margin and $b \geq 1$ is a function of $\gamma$ and other parameters of the problem, like the TS-packing number, the depth of the network, etc. Our analysis assumes that upper bounds on $1/\gamma, b$ are given, and the mistake bounds actually depend on those bounds rather than on the true correct values of those parameters. It is thus natural to seek a solution for the case that the known bounds on $1/\gamma, b$ are very loose.

In the case where only one of $\gamma, b$ is known to the learner, relatively standard doubling tricks may be used to recover the original mistake bound (up to constant factors) obtained when both are known. In this section, we show that even if both $\gamma, b$ are unknown to the learner, a mistake bound of roughly $1/\gamma^{4b}$ can be obtained. It remains open to prove or disprove that the original mistake bound $1/\gamma^b$ is achievable (up to constant, or even logarithmic factors) when both $\gamma, b$ are unknown.

The adaptive mistake bound is proven by the $\mathsf{Adap}(\mathsf{Lrn})$ algorithm given in Figure 4, where $\mathsf{Lrn}$ is the learner requiring knowledge of $\gamma, b$. Let us briefly overview algorithm $\mathsf{Adap}(\mathsf{Lrn})$. Let $\gamma^\star, b^\star$ so that $M = 1/\gamma^{\star b^\star}$ is the guaranteed mistake bound of $\mathsf{Lrn}$ when $\gamma^\star, b^\star$ are known. $\mathsf{Adap}(\mathsf{Lrn})$ maintains a guess $X$ of $M$, and for every guess $X$, it tries enough combinations of $\gamma, b$ so that $1/\gamma^b = X$. For any such combination, it runs $\mathsf{Lrn}$ and stops its execution if $X + 1$ mistakes are made. At first, we have $X = 2$. After trying all combinations of $\gamma, b$ that will be shortly defined, $\mathsf{Adap}(\mathsf{Lrn})$ updates $X := X^2$, and starts again with the new guess of $M$. For every guess $X$ of $M$, $\mathsf{Adap}(\mathsf{Lrn})$ does the following: Initialize $b := X, 1/\gamma := X^{1/b}$ and execute $\mathsf{Lrn}$ with those parameters. If $X + 1$ mistakes are made, it updates $b := b - 1, 1/\gamma := X^{1/b}$, and try again with the new parameters. $\mathsf{Adap}(\mathsf{Lrn})$ repeats this process until $b = 1$. If $X + 1$ mistakes are made with $b = 1$, it updates the guess of $M$ from $X$ to $X^2$, and repeat the process.

---



### Adap(Lrn)

**Input:** A learner Lrn requiring knowledge of $\gamma, b$.
**Initialize:** $X := 2, b := X, \gamma := 1/X^{1/b}$.

1. Execute Lrn with parameters $\gamma, b$ until it makes $X + 1$ many mistakes.
2. If $b > 1$:
   (a) Set $b := b - 1$.
   (b) Set $\gamma := 1/X^{1/b}$.
   (c) Repeat Item 1.
3. Else, if $b = 1$:
   (a) Set $X := X^2$.
   (b) Set $b := X, \gamma := 1/X^{1/b}$.
   (c) Repeat Item 1.



Figure 4: An adaptive algorithm.

**Proposition F.1.** *Suppose that* Lrn *is an algorithm with guaranteed mistake bound $M = 1/\gamma^b$ on a sequence S, assuming that $\gamma$ is a lower bound on the relevant margin definition, and $b$ is an upper bound on the relevant exponent. Then* Adap(Lrn) *described in Figure 4 has mistake bound $O(M^4)$ on S even if no such bounds $\gamma, b$ are known.*

*Proof.* Let us analyze the mistake bound of Adap(Lrn). Let $X$ be the minimal guess of $M$ such that $X \geq M$, and let us assume w.l.o.g that even for the guess $X$, all combinations of $\gamma, b$ we have tried failed (more than $X$ many mistakes are made by Lrn). Since $X \geq M$, and we begin with $b = X$ which certainly an upper bound on $b^\star$ and end with $b = 1$ which is certainly a lower bound on $b^\star$, there must be $0 \leq j \leq X - 2$ such that the two consecutive combinations

$$(1/\gamma_j, b_j) = \left(X^{1/(X-j)}, X - j\right), \quad (1/\gamma_{j+1}, b_{j+1}) = \left(X^{1/(X-j-1)}, X - j - 1\right)$$

satisfy $1/\gamma_{j+1} \geq 1/\gamma^\star$ and $b_j \geq b^\star$. This means that if Lrn is executed with $(1/\gamma_{j+1}, b_j)$, at most $1/\gamma_{j+1}^{b_j}$ many mistakes are made. We argue that a combination with values at least $(1/\gamma_{j+1}, b_j)$ will be tried in the worst case, when the guess of $M$ is changed from $X$ to $X^2$. Indeed, let the guess of $M$ be $X^2$, and let $j' = X^2 - X + j$. Then in the $j'$'th combination guess of $\gamma^\star, b^\star$, we will have $b_{j'} = X^2 - j' = X - j \geq b^\star$, and $1/\gamma_{j'} = \left(X^2\right)^{\frac{1}{X^2-j'}} = X^{\frac{2}{X-j}}$. We thus require that $X^{\frac{2}{X-j}} \geq X^{\frac{1}{X-j-1}}$, which is equivalent to $\frac{2}{X-j} \geq \frac{1}{X-j-1}$, which indeed holds for all $j \leq X - 2$, which is the required range.

Therefore, in the worst case, when $M = X^2$, Lrn will be executed with correct upper bounds on $1/\gamma^\star$ and $b^\star$ and will make on this execution, at most $\left(X^{\frac{2}{X-j}}\right)^{X-j} = X^2 \leq M^2$ many mistakes, and thus will not update the guess of $\gamma^\star, b^\star$ past this execution.

It remains to upper bound the number of mistakes made before this execution. By definition of the algorithm, for every guess $x$ of $M$, the number of mistakes made is at most $(x + 1)x \leq (x + 1)^2$. Recall that in the final guess $x$, we have $x \leq M^2$ and thus $(x + 1)^2 \leq 4M^4$. The exponential update of $X$ implies that the total number of mistakes throughout the entire execution is at most $8M^4$, as required. $\qquad\square$

# G Bounds on the TS-packing number

**Theorem G.1.** *For any $d \in \mathbb{N}^+$ and $\epsilon \leq 1/2$ we have:*

$$\max\left\{ \left(2\left\lfloor \frac{1}{2\epsilon\sqrt{d}} \right\rfloor \right)^d, d \right\} \leq \mathtt{TS}(d,\epsilon) \leq \left(\frac{1.5}{\epsilon}\right)^d.$$

*Proof.* The upper bound is the known upper bound for the standard (not totally separable) $(d,\epsilon)$-packing number (see , e.g., the lecture note [Wu and Yang, 2016]).

As for the lower bound, let us start with the first expression. Consider the $d$-unit cube inscribed in the $d$-unit ball, where the cube's sides are parallel to the axis. The vertices of the cube are thus precisely the set $\{\pm 1/\sqrt{d}\}^d$. Supppose that $\epsilon \leq \frac{1}{2\sqrt{d}}$, as otherwise the stated bound is 1, which holds trivially for $\epsilon \leq 1/2$. For every direction $i \in [d]$, we define the following $2\left\lfloor \frac{1}{2\epsilon\sqrt{d}} \right\rfloor + 1$ many hyperplanes:

$$(\boldsymbol{e_i}, k \cdot 2\epsilon), \quad \forall k \in \left\{ -\left\lfloor \frac{1}{2\epsilon\sqrt{d}} \right\rfloor, \ldots, 0, \ldots, \left\lfloor \frac{1}{2\epsilon\sqrt{d}} \right\rfloor \right\}.$$

We denote this set of hyperplanes by $D_i$. The sets $D_i$ define a grid inside the unit cube. We can choose a cell in the grid by choosing for every $i$, $j_i \in \left\{ -\left\lfloor \frac{1}{2\epsilon\sqrt{d}} \right\rfloor, \ldots, 0, \ldots, \left\lfloor \frac{1}{2\epsilon\sqrt{d}} \right\rfloor - 1 \right\}$, where $j_i$ specifies the location of the cell in the $i$'th direction: all points $x$ such that $\mathsf{sign}(\langle \boldsymbol{e_i}, x \rangle + j_i \cdot 2\epsilon) \geq 0$ but $\mathsf{sign}(\langle \boldsymbol{e_i}, x \rangle + (j_i + 1) \cdot 2\epsilon) < 0$. Since we have $2\left\lfloor \frac{1}{2\epsilon\sqrt{d}} \right\rfloor$ many choices in each direction and $d$ many directions, the number of cells is precisely $\left(2\left\lfloor \frac{1}{2\epsilon\sqrt{d}} \right\rfloor \right)^d$. By definition of the hyperplanes in the set $D_i$, each cell is a $d$-cube with side length $2\epsilon$, and thus the inscribed $d$-ball in any such cube has redius precisely $2\epsilon$. Therefore, the center of any such ball has distance at least $\epsilon$ from any hyperplane in the sets $D_i$. It is also straightforward to see that for two centers of different balls there exsits a separating hyperplane in the sets $D_i$. Therefore, the set of all centers of balls inscribed in cells in the defined grid forms a $(d,\epsilon)$-TS-packing.

It remains to prove the $d$ lower bound for any $\epsilon < 1/2$. Consider the $d$-regular simplex inscribed in the $d$-unit ball. Dentoe its vertices by $V = v_1, \ldots, v_d$. We claim that $V$ is a $(d,\epsilon)$-TS-packing. We construct for every $v_i$ a hyperplane $(\boldsymbol{w_i}, b_i)$ such that $\mathsf{sign}(\langle \boldsymbol{w_i}, v_j \rangle + b_i) > 0 \iff j = i$, while making sure that $|\langle \boldsymbol{w_i}, v_j \rangle + b_i| > 1/2$. We start with $v_1$. Consider the hyperplane $\boldsymbol{v_1}$, specified by the same values as the point $v_1$. We will show that $\mathsf{sign}(\langle \boldsymbol{v_1}, v_j \rangle) > 0 \iff j = 1$. The direction $j = 1 \implies \mathsf{sign}(\langle \boldsymbol{v_1}, v_j \rangle) > 0$ is trivial. For the other direction, let $j \neq 1$, and suppose towards contradiction that $\mathsf{sign}(\langle \boldsymbol{v_1}, v_j \rangle) > 0$. and consider the following triangle. Two of its vertices are simply $v_1$ and $v_j$. The third vertex $u$, is the intersection of the infinite line $\ell$ that intersects $v_1$ and the origin, and the infinite line that intersects $v_j$ and is also orthogonal to $\ell$. Let's calculates the triangle sides' lengthes. By assumption, $\mathsf{dist}(v_1, u) < 1$. Therefore, also $\mathsf{dist}(v_j, u) < 1$. Pythagoras' theorem now impies that $\mathsf{dist}(v_1, v_j) < \sqrt{2}$. However, this contradicts Jung's theorem, stating that the diameter of $V$ is exactly $\sqrt{\frac{2(d+1)}{d}} > \sqrt{2}$, and therefore $\mathsf{dist}(v_1, v_j) > \sqrt{2}$. To conclude, we have $\langle \boldsymbol{v_1}, v_1 \rangle = 1$ and $\langle \boldsymbol{v_1}, v_j \rangle < 0$ for all $j \neq 1$. Thus, the hyperplane $(\boldsymbol{v_1}, -1/2)$ satisfies our requirements for $v_1$. We may define a similar hyperplane for all other points in $V$. Therefore $V$ is a $(d,\epsilon)$-TS-packing for all $\epsilon \leq 1/2$. $\square$

