# OpenReview forum: "Online Learning of Neural Networks"
_NeurIPS.cc/2025/Conference — NeurIPS 2025 poster_

### Official Review · Reviewer_dAug · 2025-06-11

**Clarity:** 3
**Significance:** 2
**Originality:** 3
**Rating:** 4
**Confidence:** 3

**Summary:**

This paper proves lower and upper bounds on the problem of online learning neural networks with the sign activation function in the realizable setting. These bounds relate the problem to the "totally separable packing number", a geometric quantity. Bounds are improved for two special cases (e.g., prediction depends only on low-dimensional subspace). Finally, a known reduction from the literature is applied to also say something for the agnostic case.

**Questions:**

- How difficult is it to compute or approximate the TS-packing number? From a computational complexity perspective?
- Can Lrn use unbounded computational ressources? It seems so. Is this correct?
- The paper [Khalife, S., Cheng, H., Basu A. (2023). Neural networks with linear threshold activations: structure and algorithms (Extended version). Mathematical Programming, 2023.] shows that all linear threshold networks (which should be the same as the sign activation networks in this paper, as far as I understand) can be equivalently represented as networks with only two hidden layers. Can this be leveraged to improve Thm. 2.3, which depends exponentially on the depth? Or any other result of this paper?
- More generally, the division of a neural network into a "pipeline with two stages" reminds me of the proof strategy of the result I mention in the previous point. So maybe you should read it and potentially cite it, even if it does not yield an improvement for your results.

**Ethical Concerns:**

["NO or VERY MINOR ethics concerns only"]

**Final Justification:**

I thank the authors for their clear and extensive rebuttal. While I think that my original reservations are still somewhat present, the authors managed to convince me that, overall, their results are deep and interesting enough to justify acceptance at NeurIPS. I therefore increase my score from 3 to 4. I trust the authors that they will implement the promised changes in the final version.

**Limitations:**

adequately addressed

**Paper Formatting Concerns:**

no concerns

**Quality:**

3

**Strengths And Weaknesses:**

Strengths:
- problem studied is fundamental and important, and seems to be not well-studied so far (even though I don't know the literature and trust the authors on that).
- the paper is well-written and seems to contribute quite a few non-trivial technical results to this fundamental and important problem.
- I like that the authors put significant effort in high-level overviews of their proofs.

Weaknesses:
- I'm not sure how meaningful Thm. 2.1 is, as it mainly relate some complicated quantity (bounds on online learning) to another complicated quantity (TS-packing number). It is not immediately clear why this is a useful insight.
- Thm. 2.2 doesn't seem very surprising. It basically says "if we are in a k-dimensional subspace of R^d, then everything behaves as if we were in R^k", which seems mathematically trivial. Am I missing something here?
- For Thm. 2.3, it is not clear how realistic the assumption of "large margins everywhere" is. Is this a theortical borderline case, or is there some justification why this is interesting to study?
- Section 2.4 sounds like a fancy additional technical contribution, but in fact it is only there to remedy a weakness that was swept under the rug before, namely that gamma and b need to be known for the results in Sections 2.1 to 2.3. This should be mentioned immediately when stating the results!
- Prop. 2.5 is seems to be strong and general, but is only a minor technical contribution, as it basically follows from Hanneke et al. (2023a).
- sign activation function is not much used in practice (but I recognize it is still a fundamental case that deserves to be studied, as well-argued by the authors).

Minor (no need to discuss in rebuttal, these don't influence my evaluation):
- line 28: recieves --> receives
- line 32: Is the architecture of Phi* fixed? Known to Lrn?
- line 46: typo in "converasions"
- line 97: I don't understand what the distance between "an input point" and "a neuron" is. Also assumptions should not be "hidden" in footnotes. Later, in line 104, I again do not understand what this distance should be: (w,b) is a vector in R^{d+1}, but x_k is in R^d. Do you only compare inputs to the weight vectors, ignoring the bias? In both cases? Please make this clear.
- line 110: "Note that the difference between the upper and lower bounds is roughly quadratic in the worst case." --> what does this mean? make it more clear. does it mean that one is the square of the other (at most)?
- line 114: "not catastrophic" is very dramatic and judging wording. Can you put this into more scientific terms and justify why it is "not catastrophic"?
- line 150: this definition in prosa is quite unclear. Please provide a more rigorous definition of gamma.
- line 165: where does the b suddenly come from? Can you explain this better?
- line 193: randomized --> randomize (remove d)
- line 342: improvmnt --> improvement (insert e)

---

> ### Author Rebuttal · Authors · 2025-07-29
>
> We thank the reviewer for taking the time to carefully read and evaluate our work, and for their thoughtful and positive feedback, as well as their comments and suggestions to improve it. We will make our best efforts to incorporate their valuable suggestions in the next version of this paper. Below, we respond to the main issues raised in the review.
>
> ### Response to Weakness #1:
> We respectfully disagree with the reviewer on this weakness, for the following reasons.
> 1. In Theorem G.1 (appendix G), we provide quantitative bounds on the TS-packing number. Those bounds are proved via pure combinatorial and geometric arguments that are not directly related to learning theory. Nevertheless, by applying Theorem 2.1, those bounds translate to quantitative bounds on the mistake bound of learning neural networks online. Most importantly, the lower bound of Theorem G.1 demonstrates the unavoidable dependence of the mistake bound on the dimension, showing that additional assumptions must be made in order to obtain dimension-free mistake bounds.
> 2. The TS-packing number is not a new quantity defined in this paper. The early study of this quantity (or very similar variations of it) dates back to the 40’s, and it is an interesting quantity in its own right (see the recent thorough survey [BL24], also mentioned in Section 3.2, for more details). Our work provides even further motivation to study the TS-packing number, by proving that bounds on it immediately translate to mistake bounds for online learning of neural networks.
> 3. Characterizing the complexity of learning tasks by combinatorial (and often complicated) quantities is a long and fruitful tradition in learning theory. Perhaps the most known examples are the VC-dimension, characterizing PAC-learnability [VC71], and the Littlestone dimension, characterizing online learnability [Lit89] and PAC private learnability [ABL+22]. Other famous examples include the DS-dimension, characterizing multiclass PAC learnability [BCD+22], and both the standard and sequential fat-shattering dimensions, characterizing PAC [KS94] and online [RST15] learnability of real-valued functions, respectively. There are many more examples in the literature.
>
> ### Response to weakness #2:
> We respectfully disagree with the reviewer on this weakness as well. There is a striking difference between being in a known $k$-dimensional subspace of $\mathbb{R}^d$ and the $k$-index model (that is, the multi-index model with $k$ many directions). The difference is that in the $k$-index model, while we know that the target function depends on $k$ orthonormal directions, we do not know **what those directions are**. The challenge is to find those directions. To illustrate this, imagine that you are in the desert, and there is only one safe way out to the city. Suppose that you are being told that the way out is a straight line (that is, $1$-dimensional). While the problem is now easier than considering all possible paths, it is still much harder to find this straight line, than to simply walk on it, in case you were being told exactly where this line is.
> At a technical level, the upper bound of Theorem 2.2 is achievable thanks to the specific method used by our algorithm: It quickly identifies the boundaries between regions where the label changes such that the bottleneck (the most significant part) of the mistake bound stems from the number of such regions. The $k$-index assumption ensures that the number of such regions grows at most as $\mathtt{TS}(k,\gamma_1)$.
>
> ### Response to weakness #3:
> Theorem 2.3 provides an upper bound in terms of the minimal margin over the entire target network. Assuming that this margin is large is not necessary for the theorem to hold, but the bound indeed gets better as the margin gets larger. In other words, the assumption is required for the bound to improve over the previous bounds of Theorems 2.1-2.2.
>
> As for specific justifications, the theorem is specifically valuable for shallow networks. For such networks, the bound depends only polynomially on the margin’s inverse, which is reasonably tight, by the well known lower bound for learning linear classifiers mentioned after the statement of the theorem (lines 157-158).
>
> To illustrate this, let’s consider a prototypical case of a network with a single hidden layer and one output neuron. In such a network, the difference between the set of neurons in the first layer considered in Theorem 2.2 and the entire set of neurons is only the output neuron. Nevertheless, the bound of Theorem 2.3 in this case depends only polynomially on the margin’s inverse, and has no dependence on the input dimension. Thus, it suffices that the margin in the output neuron is not extremely small for the bound of Theorem 2.3 to be much better than the bound of Theorem 2.1.
> We will add this example to the next version of the paper. Thank you for bringing this to our attention.
>
> ### Response to weakness #4:
> We agree that it will be better to clarify earlier that those parameters should be known to the learner in order to obtain the exact bounds of Theorems 2.1-2.3. Thank you for bringing this to our attention. We would also like to stress that the degradation caused by not knowing the parameters in advance is only polynomial, as stated in Theorem 2.4.
>
> ### Response to weakness #5:
> We agree that Proposition 2.5 is strong and general, **and** only a minor technical contribution – those two statements do not contradict each other. The main technical contributions of the paper are indeed for the realizable case. This is explicitly stated right at the beginning of the paper (lines 31-37).
>
> ### Response to weakness #6:
> We agree with this weakness and with the reviewer’s reservation inside the parentheses. We will add an open question on studying a similar setup with other activations to the next version of the paper.
>
> ### Response to minor weaknesses:
> At the reviewer's request, we will refrain from discussing those weaknesses. We just want to thank the reviewer for their detailed evaluation, and for pointing out typos and unclear points. We will make sure to address and fix these issues in the next version of the paper.
>
> ### Response to question #1:
> This is a very interesting question for future research, and we will add it to the relevant paragraph, starting at line 379. The best approximation of the TS-packing number we know is the bounds stated in Theorem G.1 of the appendix. We are not aware of any bounds on the computational complexity of calculating a better approximation of the TS-packing number.
>
> ### Response to question #2:
> Yes, we did not restrict the computational complexity of the learner. Designing computationally efficient algorithms for online learning of neural networks is an interesting direction for future work. We have mentioned it in the “future work” section.
>
> ### Response to questions #3 and #4:
> Thank you very much for bringing this relevant paper to our attention. We indeed were not aware of it. The result you have mentioned unfortunately does not seem to directly improve the bound of Theorem 2.3 (or any other bound), since the process of converting a net of depth L to a net of depth 3 (that is, one with 2 hidden layers) might decrease the margin. However, this might be a possible interesting approach towards improving it. We will definitely add this to Section 5.2, where we discuss the open question of improving the exponential dependence on the depth in Theorem 2.3 (or proving that it is impossible).
>
> It is also true that our proof strategy of Theorem 2.1 has some similarities with the proof strategy of the result that you have mentioned. We will relate to that matter as well in the next version of this paper.
>
> ### References:
>
> [VC71] Vladimir N. Vapnik and Alexey Ya. Chervonenkis. On the uniform convergence of relative frequencies of events to their probabilities. Theory of Probability & Its Applications, 16(2):264–280, 1971.
>
> [ABL+22] Noga Alon, Mark Bun, Roi Livni, Maryanthe Malliaris, and Shay Moran. Private and online learnability are equivalent. ACM Journal of the ACM (JACM), 69(4):1– 34, 2022.
>
> [BCD+22] Nataly Brukhim, Daniel Carmon, Irit Dinur, Shay Moran, and Amir Yehudayoff. A characterization of multiclass learnability. In 2022 IEEE 63rd Annual Symposium on Foundations of Computer Science (FOCS), pages 943–955. IEEE, 2022.
>
> [KS94] Michael J Kearns and Robert E Schapire. 1994. Efficient distribution-free learning of probabilistic concepts. J. Comput. System Sci. 48, 3 (1994), 464–497.
>
> [RST15] Alexander Rakhlin, Karthik Sridharan, and Ambuj Tewari. Online learning via sequential complexities. J. Mach. Learn. Res., 16(1):155–186, 2015.
>
> [BL24] K´aroly Bezdek and Zsolt L´angi. On separability in discrete geometry. arXiv preprint arXiv:2407.20169, 2024.

---

> > ### Comment · Reviewer_dAug · 2025-08-01
> >
> > I thank the authors for their clear and extensive rebuttal. While I think that my original reservations are still somewhat present, the authors managed to convince me that, overall, their results are deep and interesting enough to justify acceptance at NeurIPS. I therefore increase my score from 3 to 4. I trust the authors that they will implement the promised changes in the final version.

---

### Official Review · Reviewer_A2P8 · 2025-06-24

**Clarity:** 3
**Significance:** 3
**Originality:** 3
**Rating:** 5
**Confidence:** 3

**Summary:**

This paper studies online learning for the class of feedforward neural networks of a fixed architecture (that maps the unit ball to $Y=\{\pm 1\}^k$ for some finite $k$) with the sign activation function. In the realizable setting, where the sequence presented by an adversary respects the labels assigned by some neural network in the class, the first result in the paper (upper bound in Theorem 2.1) shows that there exists a learning algorithm, that makes at most $\tilde{O}(TS(d, \gamma_1)/\gamma_1^2)$ mistakes on any sequence $S=\{x_1,\dots,x_T\}$ labeled by some target network $N$; here, $\gamma_1$ is the smallest absolute value (margin) of a neuron (pre-activation) in the first layer of $N$ over all the examples in $S$, $d$ is the input dimension, and $TS(d, \epsilon)$ denotes the totally-separable packing number of the unit ball, which is the size of the largest set such that there exists a hyperplane that separates every pair of points in the set, and is also at least $\epsilon$ away from every point in the set. Complementary to this, the lower bound in Theorem 2.1 shows that for every learner, there exists a network and sequence of examples for which the number of mistakes is at least $\Omega(TS(d, \gamma_1)+(1/\gamma_1^2))$.

$TS(d, \epsilon)$ depends poorly on the dimension $d$ (sometimes being exponential in $d$ for small $\epsilon$)--- thus, to obtain dimension-free bounds, the authors consider two restrictions of the learning setting. The first setting considers the class of sign activation neural networks where the overall function computed by the neural network can be written as some function of the linear projection of the input onto a $k$-dimensional subspace. For this class, denoted as the multi-index model, the authors show (Theorem 2.2) a mistake upper bound of $\tilde{O}(TS(k, \gamma_1)/\gamma_1^2)$. The authors show that $TS(k, \gamma_1) \le (1.5/\gamma_1)^k$, so for small values of $k$, this mistake bound is $1/poly(\gamma_1)$.

The second setting that the authors consider is that where every neuron pre-activation in the *entire* neural network has absolute value at least $\gamma$ over all the examples in the sequence $S$. For this setting, the authors show an upper bound of $\tilde{O}(\log(|Y|)/\gamma^{\Theta(L)})$, where recall that $Y$ is the label space.

The algorithms above assume knowledge of $\gamma_1$ or $\gamma$: the next results in the paper consider the setting in absence of this knowledge. Theorem 2.4 shows the existence of an algorithm that does not require this knowledge, and attains a mistake bound that is only polynomially-worse than the mistake bound of the algorithm that knows this information. Lastly, the authors touch upon the agnostic setting (Propostiion 2.5)---using a reduction from agnostic to realizable learning for multiclass online classification established by Hanneke et al. 2023a.

**Questions:**

1) Maybe I am missing something obvious, but with regards to the footnote at the bottom of page 4, why does the lower bound from Theorem 2.1 trivially hold in the multi-index setting? Is it clear that the lower bound instance from Theorem 2.1 is a multi-index model?

2) With regards to pruning: Is the following conclusion accurate given the results in your paper: For any neural network, there exists a small subset of neurons in the first layer, such that, if we retain only these neurons and get rid of all the other parameters of the network, then one can get a near-optimal mistake bound for the online learning game played on examples labeled by the unpruned network?

3) I was trying to parse through the proof of the upper bound Theorem 2.1 in Appendix A, and the final form of the labelling function $L_G$ that the learner uses for the expert in Figure 3 was not clear to me. Could you please spell this out, and also point me to where it gets defined in the proof in Appendix A?

4) Consider the class of neural networks of a fixed architecture (say with sign activations, and a single output neuron)---this class has some Littlestone dimension $d$. Standard realizable online learning says that $d$ is the optimal mistake bound for realizable online learning for deterministic algorithms. Is it fair to say that the upper and lower bounds in Theorem 2.1 are in some sense, *instance-specific* bounds for every target function from the class? That is, is it clear that the upper bound in Theorem 2.1 is uniformly always smaller than the Littlestone dimension of the class? Also, it must be the case that the lower bound is smaller than the Littlestone dimension (because of SOA achieving this bound always), right? Please let me know if what I am asking does not make sense.

**Ethical Concerns:**

["NO or VERY MINOR ethics concerns only"]

**Final Justification:**

The authors' response helps clarify some of my confusions. Assuming that they will address them in the revision, I will maintain my positive review of the paper, and remain in favor of accepting the paper.

**Limitations:**

yes

**Quality:**

3

**Strengths And Weaknesses:**

The paper is generally very well written, and does a good job of motivating the problem and also situating the work amidst relevant literature on online learning of neural networks. The proof sketches provide sufficient insights with regards to the technical tools used. The mistake bounds presented in the paper in the general setting are near-optimal. The algorithm used for the general setting is a nice adaptation of the standard multiplicative weights algorithm with experts; the class of experts constructed in the paper, and the essential arguments that make the experts algorithm work---namely 1) class of experts is small and 2) some expert is accurate--- are interesting to see. The techniques required to make the general-purpose algorithm work in the more restrictive settings are non-trivial, especially the setting that assumes margin over all the neurons. Lastly, the upper bounds can be be viewed as instantiations of the network-pruning/lottery hypothesis principle, which is quite a curious result.

As for weaknesses: it would be great if the definition of the precise online learning game considered in the paper (and the definition of the class of neural networks being considered) is included in the main paper, right before stating the results, in a preliminaries section. I understand the page limit restrictions, but hopefully, the extra page after paper decisions will help with this. As written right now, in order to fully understand all the terms in the statement of Theorem 2.1, I had to glance through Appendix A to understand some of the definitions and the precise problem setting.

---

> ### Author Rebuttal · Authors · 2025-07-29
>
> We thank the reviewer for taking the time to carefully read and evaluate our work, and for their thoughtful and positive feedback, as well as their comments and suggestions to improve it. We will make our best efforts to incorporate their valuable suggestions in the next version of this paper. Below, we respond to the main issues raised in the review.
>
> ### Response to weakness:
> We completely agree that some of the terms are only defined in the appendix (as noted in the beginning of Section 2). If the paper is accepted and an extra page is given for the camera-ready version, we will absolutely do our best to include all the key definitions in the main body.
>
> ### Response to question #1:
> Consider the $k$-index model (that is, multi-index model with $k$ signals). One can apply the lower bound of Theorem 2.1 with dimension $k$ to the $k$-index model with dimension $d$: we use the instances from the lower bound with dimension $k$, as $d$-dimensional instances where every entry after the $k$’th is zero. Therefore, the target function depends only on the first $k$ directions, and the lower bound matches the statement of Theorem 2.1 with dimension $k$.
>
> ### Response to question #2:
> Not exactly. Our work uses pruning in two kind of different ways. In Theorems 2.1 and 2.2, we indeed mostly rely on pruning the first layer of the network. Therefore, with respect to those results, your conclusion is correct, with the two following reservations:
> 1. The subset of neurons remaining in the first hidden layer (after pruning it) might be as large as $\mathtt{TS}(d,\gamma_1)$, which might not be small, and could even be exponential in $d$ for small $\gamma_1$, as demonstrated in Theorem G.1 (in Appendix G). Furthermore, this quantity heavily depends on the input, not just on the network, by the definition of $\gamma_1$.
> 2. After pruning, while we can reconstruct the original networks’ labeling based only on the remaining neurons, it is not as simple as just pruning the unimportant neurons and recalculating the value of the pruned network. In more detail, while the pruning process preserves the boundaries between regions with different labels, it loses grasp of **which** label is given to each region, and this knowledge must be reconstructed without relying on the original architecture of the network. We included an explanation in this spirit in Section 3.3, but we will add the above more detailed explanation to the next version of the paper.
>
> However, the pruning used in Theorem 2.3 is more similar to your description: it removes a portion of the network’s parameters (and the size of this portion increases with $\gamma$), and the new, pruned network calculates the same labels as the original network. The difference from your description, is that the depth of the network is preserved, and we just have less neurons in each layer. We will also add this explanation to the next version of the paper.
>
> ### Response to question #3:
> You are absolutely correct that we did not explicitly initialize the labeling function $L_G$ used by the expert in the context of Theorem 2.1. The reason is that the initialization has no effect on the mistake bound of the expert (up to a constant factor). As explained in the second item of our response to question #2, the labeling function needs to be reconstructed from scratch. This seemingly pricey reconstruction has no effect on the expert’s mistake bound, since the bottleneck (that is, the more difficult part) is to identify the boundaries between regions with different labels. That bottleneck is referred to as “mistakes of the first type” and denoted by $M_1$ in Section B.1 of the appendix. Mistakes caused by an incorrect labeling function are what we call  "mistakes of the second type”, denoted by $M_2$. We show in Lemma C.7 in the appendix that even when reconstructing the labeling function from scratch, we have $M_2 \leq M_1$ and thus the total mistake bound $M_1+M_2$ is at most $2 M_1$.
>
> After clarifying that, we agree that adding an explicit initialization of the labeling functions to arbitrary values (like fixed $1$) will be more accurate and clear, and we will add such an initialization to the next version of the paper. Thank you for bringing this issue to our attention.
>
> ### Response to question #4:
> As mentioned in lines 80-82, even in a very minimal setting of a single input neuron and a single output neuron, any learner can be forced to make an unbounded number of mistakes in the online setting, unless more assumptions are made. This implies that even for the class defined by this simple architecture, the Littlestone dimension is infinite. We are able to obtain finite mistake bounds only when placing more assumptions, involving also the input, not just the class of networks.
>
> However, if one wishes to discuss online learning of neural networks in terms of the Littlestone dimension of the relevant class, a possible approach is to use the concept of a partial hypothesis class defined in [AHHM22]. This framework allows to define a hypothesis class as a set of partial functions, enforcing restrictions on the input sequence, such as large margin. We briefly mentioned that in footnote 5 in the appendix.
>
> ### References:
>
> [AHHM22] Noga Alon, Steve Hanneke, Ron Holzman, and Shay Moran. A theory of pac learnability of partial concept classes. Annual Symposium on Foundations of Computer Science (FOCS), 2022.

---

> > ### Comment · Reviewer_A2P8 · 2025-08-01
> >
> > I thank the authors for their detailed response. I would encourage them to particularly address Question 3 in the revision. I will maintain my score and evaluation.

---

### Official Review · Reviewer_8nh1 · 2025-06-27

**Clarity:** 4
**Significance:** 3
**Originality:** 3
**Rating:** 5
**Confidence:** 4

**Summary:**

This paper gives mistake upper and lower bounds for the problem of online learning of neural networks that have sign functions. It seems that the important parameter is the totally separable packing number (TS) which depends on the dimension of input and $\gamma_1$ which is the margin of the neuron’s of the first layer from the sample. The authors demonstrate a mistake bound that scales with the $TS(d, \gamma_1)$. This dependence can be exponential in the dimension $d$. Given this, they consider a different multi-index setting, where the labeling of data only depends on some $k <<d$ unknown dimensions. In this case, they show the learner enjoys a mistake bound that only scales with $TS(k,\gamma_1)$. Another restricted setting is where every neurone in every layer has margin $\gamma$. The authors show that in this setting the dependence on the dimension can be removed but an exponential dependence on the depth appears. I like the idea used here where in the space of first layer’s neurons each sample is considered as a hypothesis that gives a value to the neurons. Then it is possible to define a probability distribution over the neurons which is proportional to the weights with which the neurons are connected to the next layer/output. Using uniform convergence we can guarantee a small set of neutrons exists that approximates the output with an accuracy that does not change the sign, thus the output remains the same.

For the mistake bound a lower bound is also provided which is not tight an has a gap of approximately $1/\gamma_1^2$ in some cases.

The authors also make progress in relaxing the knowledge of the margin parameter $\gamma_1$ and the dimension. I think this is an important part that is worth exploring further because $\gamma_1$ is a value that strongly depends on the entire sample.

All the upper bounds are derived for the agnostic case using known agnostic to realizable reduction techniques.

**Questions:**

The results are considered for the input lying in the unit ball. What is the dependence on the radius of the input? Similar to the linear classifiers where we get a mistake of $R^2/\gamma^2$ for radius $R$ and margin $\gamma$, is it possible that the dependence comes in the form of a value $O(R/\gamma_1)$ in the packing number? Does generalizing the unit ball require different approaches or do the authors  focus on the unit ball for the simplicity of the notation?

How would one attemp other activation functions? Like if we have ReLU, what happens to the regions? There is a bit of a mention in related work that we can replace the activation by Lipschitz activations with a Lipschitz parameter that dependens on the margin everywhere parameter. But what if we have some layers with smooth activation functions, probably up to last layer, and then have sign functions. Can we reason about how the combined layers separate the space in regions and try to find the set of experts we need? It seems that still the parameter $\gamma$ could be controlled by the weight of the layers and the Lipschitzness of the layers. Would this then be different than the prior work mentioned or will it give new understandings? This will potentially depend on the smoothness and weight of different layers and the margin of the first layers. To me it seems that we can kind of separate the first layers and then conclude a margin for it, which depends on its depth and weight of the vectors. Then the same analysis for the rest and output layer can be applied.

**Ethical Concerns:**

["NO or VERY MINOR ethics concerns only"]

**Final Justification:**

I am happy to keep my score. The authors responded well to my questions. I also believed that the questions raised are best suited for future work and the paper is sufficient in its own. The reason I asked the question is that the paper provoked them, which makes me believe that the problem is interesting.

**Limitations:**

I believe the main important limitations are already mentioned in the paper. Please see Questions and Strengths and Weaknesses section for more comments.

**Paper Formatting Concerns:**

None.

**Quality:**

3

**Strengths And Weaknesses:**

The paper is well-written, clear and concise. All the ideas are clearly explained and elaborated as much as possible. For the technical parts, please see the summary. In general, I enjoyed the technical part and the attempts to make progress with different assumptions.

I think the question of improving the exponential dependence on depth is an important question because in the supervised classification setting, the depth usually appears linearly. It seems to me that the exponential dependence here still does not come from the nature of the problem, rather than from the technique to approximate a function, i.e., neural network, with a different function that depends on less parameters and is easier to learn. So it is really not clear if the dependence is necessary, as pointed out by the authors too.

I also think knowing having the knowledge of the $\gamma$ is a very important challenge. I understand in a lot of online learning settings, it is easy in to relax this notion by simulating the learner and doubling the parameters iteratively. In this setting I also think it is a very important parameter because of its dependence on the sample. I however think this may require more efforts and can be addressed in future work. I also think that providing a bound that removes this requirement, although at an exponential price, is still important, which the authors have provided.

---

> ### Author Rebuttal · Authors · 2025-07-29
>
> We thank the reviewer for taking the time to carefully read and evaluate our work, and for their thoughtful and positive feedback, as well as their comments and suggestions to improve it. We will make our best efforts to incorporate their valuable suggestions in the next version of this paper. Below, we respond to the main issues raised in the review.
>
> ### Response to "Strengths and weaknesses”:
> We thank the reviewer for the positive and encouraging feedback. We want to make sure that the contribution of Section 2.4 is clear. The price of removing the requirement for prior knowledge of the problem’s parameters (including $\gamma$) is polynomial, not exponential: the bound we prove is $O(M^4)$, where $M$ is the bound obtained when knowing the parameters in advance. We agree, of course, that even though the upper bound on the “price” that we prove is only polynomial, it will be interesting to improve it (or to show that it is impossible), as also mentioned in the open questions section.
>
> ### Response to question on input radius:
> Yes, the same approach can be used for any radius R, and we use the unit ball since it is a simplifying assumption that does not actually simplify the problem. Let’s consider linear classifiers as a simple example, as also done in the review. Any input of radius $R$ and margin $\gamma$ can be shrinked such that it fits into the unit ball, while its labeling does not change. The only difference is that the shrinkage causes the margin to decrease from $\gamma$ to $\gamma’ = \gamma/R$. For linear classifiers, this translates to a mistake bound of $1/\gamma’^2 = 1/(\gamma/R)^2 = R^2/\gamma^2$, as in the original problem. The same shrinking can be done in our setup, and the margin decreases proportionally to the original radius.
>
> ### Response to question on other activations:
> Regarding ReLU activation, we unfortunately do not see how our approach can be directly applied to the ReLU activation. A main reason is that our approach uses the classic Perceptron binary online learner as a building block. Roughly speaking, since the sign activation is also binary, this allows us to converge to the correct regions induced by the activation outcomes of the "important" neurons. Obtaining similar results for the ReLU activation might require a completely different approach, and this is an excellent direction for future work. We will add it to the “future work” section. Thank you for bringing this to our attention.
>
> Regarding the second part of the question, this sounds like a nice direction for future work. If we understood the reviewer’s intention correctly, they wonder whether we can relax some of the assumptions by replacing the activations in the first layers from sign to a smooth and Lipschitz activation, and then having control over the margin by some assumptions on the weights, rather than a direct margin assumption. It sounds reasonable, but might require further work in order to be formally proven.

---

> > ### Comment · Reviewer_8nh1 · 2025-08-07
> >
> > I thank the authors for their rebuttal. I am happy that they think the question proposed is worth including in future work.

---

### Official Review · Reviewer_J2Nn · 2025-07-02

**Clarity:** 4
**Significance:** 3
**Originality:** 4
**Rating:** 5
**Confidence:** 3

**Summary:**

Authors study deriving mistake bounds of NNs with sign activations in the online learning setting.
The study considers both the realizable and agnostic cases, and introduces multiple bounds with varying assumptions. Specifically, authors first start with acceptably tight bounds that depend on $TS(d, \epsilon)$ terms that are exponential in $d$ (input dimensionality) when $\epsilon$ is small. Then, they move away from $d$ by projecting the input to a $k<<d$ dimensional space, and operate on the lower dimensional manifold that input is assumed to be in. This helps the authors to recover a bound with $TS(k, \gamma_1) \leq (1.5 / \gamma_1)^k$ that is $poly(1/\gamma_1)$. Then, the authors extend the $\gamma_1$ treatment to all neurons and derive a logarithmic bound that has an exponential dependency to the NN depth.

**Questions:**

I have 2 main questions, both stem from my comments above. Regardless, I believe the work is strong and my questions are more towards gaining insights from the authors. If needs to be deprioritized to focus on critical questions from the other reviewers, I don't plan to change my ratings.

1) Can you briefly expand on how well your analyses would generalize to say a ReLU network? What would need to change in your approach to acquire similar bounds, and why wasn't it studied in this work?

2) A crude way to interpret these findings is that NNs can be learned online with reasonable regret when either intrinsic dimensionality of data is low /pattern can be learned by a pruned NN / or the NN is sufficiently shallow (which is somehow complementary to low data complexity). At what point these limitations become too strict that fitting linear model with logarithmic regret is more favorable?

**Ethical Concerns:**

["NO or VERY MINOR ethics concerns only"]

**Final Justification:**

Very strong paper, hence I keep my score at 5. A 6 would be findings that are groundbreaking and with immediate practical implications.

**Limitations:**

Yes

**Quality:**

4

**Strengths And Weaknesses:**

This is a strong paper that studies a fairly underexplored are from a theoretical perspective. It was as refreshing read as many existing works on online learning w/ NNs are highly empirical and they rely on questionable heuristics. Characterization of the problem and the use of pruning seem to be on point, and being able to move away (w/ assumptions) from depending to input dimension seems novel. The findings also align with empirical analyses and common insights on NN learning dynamics as the 3rd bound is exponential in L. The paper is also well-written and easy to follow - I especially appreciate the inclusion of Section 4 to the main body.

Most of my feedback is around organization and emphasis of the points. As the authors also emphasize, the findings do not immediately translate to practice as the study mainly focuses on sign activated NNs. The authors briefly mention the learnings would generalize to popular activation functions, but I believe it needs to be expanded upon. I also like the implied connections to pruning and autoregressive learning, however, it would be great if the authors provide more insights on those connections and propose how their findings can benefit these existing techniques. Finally, given the high quality of the rest of the sections (and appendix), open questions section needs to be rewritten to convey the authors' vision on the domain better.

---

> ### Author Rebuttal · Authors · 2025-07-29
>
> We thank the reviewer for taking the time to carefully read and evaluate our work, and for their thoughtful and positive feedback, as well as their comments and suggestions to improve it. We will make our best efforts to incorporate their valuable suggestions in the next version of this paper. Below, we respond to the main issues raised in the review.
>
> ### Response to weaknesses:
> We thank the reviewer for their suggestions to clarify and expand on the implications of our work to other activation functions, pruning, and autoregressive learning. We will do our best to make sure that those implications are reflected in the text in a clear and detailed way.
> We will also make another pass on the open questions section and will do our best to draw a more concrete vision for future work.
>
> ### Response to question #1:
> Unfortunately, we do not see how our approach can be directly applied to the ReLU activation. A main reason is that our approach uses the classic Perceptron binary online learner as a building block. Roughly speaking, since the sign activation is also binary, this allows us to converge to the correct outcomes of the activations of the "important" neurons, which are necessary to predict the final outcome of the network. Obtaining similar results for the ReLU activation might require a completely different approach, and this is an excellent direction for future work. We will add it to the “future work” section. Thank you for bringing this to our attention.
>
> ### Response to question #2:
> The answer may heavily depend on the parameters of the problem. For example, in many online learning settings the horizon ($T$) is considered to be very large with respect to other parameters of the problem. In such cases, even if  the parameters of the problem (like dimension, margin, or network’s depth) cause our realizable mistake bounds to be very large, it might still be better than even a logarithmic dependence on $T$.
>
> On the other hand, if $T$ is not extremely large, then in cases where a linear model with logarithmic regret is reasonably close to realize the data, and yet for a neural network to completely realize it we need really bad parameters (like a really small margin), the overall number of mistakes made by the linear model might indeed be smaller.

---

> > ### Comment · Reviewer_J2Nn · 2025-07-31
> >
> > I thank the authors for their response.

---

### Decision · Program_Chairs · 2025-09-17

**Decision:**

Accept (poster)

**Comment:**

This paper addresses the problem of deriving mistake bounds for online learning of neural networks with sign activation functions. Following the initial reviews, the authors could convincingly address most reviewer concerns, also leading to score increases. Reviewers agree that the paper presents an interesting and valuable contribution to the field, and, from my perspective, the manuscript is a nice theoretical take on a somewhat underexplored (as also mentioned by J2Nn) branch of online learning of NNs. I am recommending acceptance of this manuscript, and encourage the authors to work-in the requested changes from the reviewers, as well as their own clarifications from the rebuttal phase.